# SCALING OPEN-ENDED REASONING TO PREDICT THE FUTURE

## ABSTRACT

High-stakes decision making involves forward-looking reasoning under uncertainty. In this work, we train language models to make predictions on open-ended questions about the future. To scale up training data, we continually synthesize novel forecasting questions from global events reported in daily news, using a fully automated, careful curation recipe. We train the Qwen3 thinking models on our dataset, `OpenForesight`. To prevent leakage of future information during training and evaluation, we use an offline news corpus, both for data generation and retrieval in our forecasting system. Guided by a small validation set, we show the benefits of retrieval, a supervised finetuning phase, and an improved reward function for reinforcement learning (RL). Once we obtain our final forecasting system, we perform held-out testing between May to August 2025. Our specialized model, `OpenForecaster` 8B, matches much larger proprietary models, with our training improving the accuracy, calibration, and consistency of predictions. We find calibration improvements from forecasting training generalize across popular benchmarks. We will open-source our models, code, and data to make LLM based forecasting research broadly accessible.

## 1 INTRODUCTION

Every day, people navigate decisions under high uncertainty due to incomplete evidence and competing hypotheses. The highest-stakes choices are inherently forward-looking: governments set policy while anticipating macroeconomic and geopolitical shifts; investors allocate capital amid market and regulatory uncertainty; individuals choose careers as technologies evolve; and scientists pursue research directions in search of the next breakthrough. Decades of work (Tetlock et al., 2014) on human forecasting shows that while prediction is hard and skill varies widely, it is possible to train humans to become better forecasters. Some "superforecasters" consistently outperform peers. While there is a ceiling to predictability in social systems (Franklin, 1999), we do not yet know where that ceiling lies in the real world.

If trained at scale for forecasting world events, language models may enjoy structural advantages over humans: they can ingest and synthesize vast, heterogeneous corpora across thousands of topics; and update predictions rapidly as new text arrives. Just like language models now show superhuman reasoning on some exam-style math and coding problems (OpenAI, 2025), in the future, language model forecasters may be able to come up with possibilities that humans miss. So in this work, we study:

*How can we train language models to better forecast open-ended questions?*

**Scaling training data for forecasting.** As forecasting is hard for humans, detailed and correct reasoning traces for forecasting are difficult to obtain. Fortunately, recent success in Reinforcement Learning (RL) for language models enables training with just the eventual outcome of the question. Further, the static knowledge cutoff of LLMs enables a unique opportunity: events that resolve after the cutoff are in the future for the model. Even then, sourcing questions at scale for training forecasting abilities has a few key challenges. First, waiting for events to resolve is too slow as a feedback loop for training. Second, prediction markets–the primary source for existing forecasting questions–mostly consist of binary yes or no questions. As there is a 50% chance of success on these questions even with incorrect reasoning, they make for noisy rewards.

Thus, we synthesize open-ended forecasting questions like "Who will be confirmed as the new prime minister of Ukraine on 17 July 2025?" using global news, which covers a large number of salient events every day. To avoid shortcuts and ensure quality, we carefully curate data through

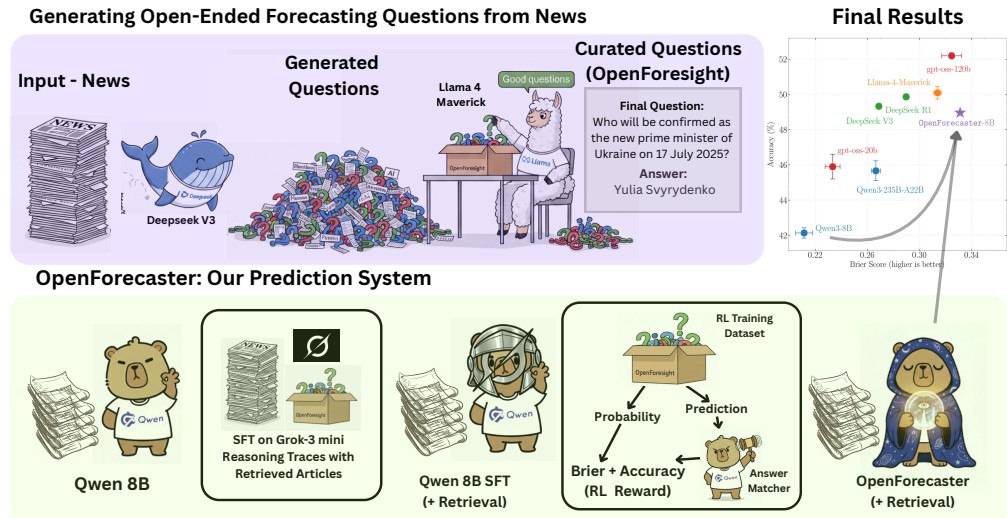

Figure 1: A summary of our methodology for training language models for open-ended forecasting.

filtering. Our recipe for creating training data is entirely automated and scalable, with one language model extracting events from news articles to generate questions, and a different model filtering and rewriting questions. For this work, we use this recipe with 250,000 articles up till April 2025, to create `OpenForesight`, a dataset of 60,000 open-ended forecasting questions for training. To grade responses to open-ended questions, we use model-based *answer matching* consistent with frontier benchmarks like the Humanity's Last Exam (Phan et al., 2025).

**Ensuring we truly improve forecasting.** We take extensive measures to avoid the leakage of future information during training and evaluation. First, we do not use online search engines for sourcing news, as they have unreliable date cutoffs due to dynamic updates to documents and search ranking (Paleka et al., 2025a). Instead, we use the CommonCrawl News corpus, which provides static, monthly snapshots of global news. Second, we use open-weight Qwen3 models, only training on events until April 2025 when the model weights were released, and performing final tests between May to August 2025. Finally, we do not observe performance on the test set until the very end. Our test set is composed of diverse news sources, different from the ones used in training, to ensure we are not just learning distributional biases of the training data.

**Validating design choices for LLM Forecasting Systems.** We start from Qwen3 (Yang et al., 2025) 4B and 8B models with thinking enabled. We perform all ablations on a small validation set, using a separate source from our test set. We use dense retrieval with the Qwen3-8B Embedding model to provide forecasters relevant chunks from our offline news corpus, and see large improvements. This is despite a cautious approach of only retrieving articles until *one month* before the question resolution date to avoid leakage. We find an initial distillation step on reasoning traces from a larger model with 10,000 questions significantly improves both initial accuracy, and pass@k accuracy, with the latter being an indicator of potential for Group Relative Policy Optimization (GRPO) (Shao et al., 2024) training. For GRPO, we propose optimizing both accuracy, plus an adaptation of the brier score for open-ended responses (Damani et al., 2025). Ablations show rewarding accuracy alone hurts calibration, while optimizing only the brier score hurts exploration on hard questions.

**Final results**. In Section 6, we show RL training on `OpenForesight` yields large improvements in accuracy and calibration on our held-out test set of open-ended forecasting questions about global events. Our specialized 8B model matches much larger proprietary models. We observe calibration from forecasting training generalizes across multiple downstream benchmarks.

**Outlook.** Forecasting systems, if realized responsibly, could transform policy making, corporate planning, and financial risk management by providing rigorous probabilistic predictions (Tetlock, 2017). To promote forecasting research, we will open-source our models, code and data.

## 2 RELATED WORK

**Forecasting World Events.** Much prior work in Machine Learning and Statistics has focused on forecasting numeric data, for diverse time-series data (Box & Jenkins, 1976) like weather (Richardson,

1922), econometrics (Tinbergen, 1939) or finance (Cowles, 1933). Our work, however, focuses on the prediction of discrete world events, with both questions and answers described in natural language, also called *judgemental forecasting* (Tetlock & Gardner, 2016), which we will refer to as just *forecasting* for brevity. In prior work on evaluating language models for forecasting (Zou et al., 2022; Karger et al., 2024), questions are primarily sourced from prediction markets like Metaculus, Manifold, and Polymarket. Prediction markets, which have rapidly grown in popularity over the last few years, provide a platform for online participants to register predictions with fake or real money on questions like "Will Donald Trump win the US Presidential Election in 2024?", which mostly have binary, yes or no, outcomes.

**Evaluating LLMs for Forecasting.** Forecasting benefits from recent knowledge (before the event resolves), so LLM forecasting work (Zou et al., 2022; Halawi et al., 2024) provides relevant retrieved articles to models (Lewis et al., 2020) often obtained via web-search APIs. Paleka et al. (2025a) discuss pitfalls of LLM forecasting evaluations, including leakage of outcomes from online search in backtests, and distributional biases of prediction market questions. To avoid these issues, we focus on forecasting questions generated from an offline, reliably dated collection of global news. This is consistent with Jin et al. (2021), who used humans to create questions, while Dai et al. (2024) showed this process can be automated with LLMs. However, their questions pre-define a few outcomes to choose from, while Guan et al. (2024); Wang et al. (2025) evaluate open-ended forecasts. We move beyond evaluations, to train models for open-ended forecasting.

**Reinforcement Learning for LLMs.** Shao et al. (2024) proposed *Group Relative Policy Optimization* (GRPO), an RL algorithm that only uses outcome rewards. This approach has been highly successful in training LLMs to *reason* about well-specified coding (Jain et al., 2024) and exam-style questions across domains (Phan et al., 2025). Even before this, Halawi et al. (2024) proposed training language models for forecasting, by finetuning the model on its own chain of thought traces that led to correct predictions for prediction market questions resolving before the evaluation period begins. Recently, Damani et al. (2025) train models to accurately verbalize their uncertainty, by optimizing a joint reward of accuracy and calibration scores with GRPO. Turtel et al. (2025a) apply this to binary (yes or no) forecasting questions from prediction markets. Our work departs in showing how to synthesise large-scale open-ended questions about global events to train models that reason about the future.

## 3 OPEN-ENDED FORECASTING

**Motivation.** The forecasting task we study is *open-ended* in two key ways: 1) It allows expressing arbitrary natural language questions 2) It may not have a structured outcome set, unlike numeric or categorical predictions. This differentiates it from both time-series forecasting, and prediction markets. For example, prediction markets are dominated by binary (yes/no) or multiple choice questions. While this design is easy to score, it restricts to forecasting questions with a known, fixed set of outcomes. However, the most foresight often lies in predicting the unexpected, or when a large number of possibilities could occur. The most important questions to forecast—such as scientific breakthroughs, geopolitical shocks, or technological disruptions—often emerge as *unknown unknowns*: possibilities not anticipated, and hard to enumerate. Thus, in this work, we focus on training models to make open-ended predictions like "Which company will the US Government buy a >5% stake in by September 2025?". Such questions require exploration and imagination, rewarding the creation of completely new hypotheses that turn out to be correct, rather than just distributing probabilities over a known set of outcomes.

**Background.** LLM weights are frozen after training, especially when the weights are released openly. Any event that happened between the last date in their training corpus is in the future for the LLM. This provides a time window from which to collect questions for training models to reason about future events. Similarly, their evaluation involves testing on questions resolving after the cutoff date of the training data, called *backtesting* (Tashman, 2000). While prior work has relied on prediction market questions as training data, this has three key problems. First, the questions are created by humans, which makes them low in number (Paleka et al., 2025a). This becomes a bottleneck for scaling training data, which has been an essential component in the success of LLMs (Kaplan et al., 2020; Lu, 2025). Second, a large majority of questions have binary outcomes, which creates a 50% baseline success rate. This means even incorrect reasoning has a high chance of being reinforced. This leads to noisy rewards in outcome-based RL. Third, prediction markets overrepresent US politics, with individual platforms emphasizing niches: Polymarket (crypto),

Metaculus (technology), Manifold (personal life), and Kalshi (sports) (Paleka et al., 2025a). These limitations motivate us to explore alternate ways to create forecasting questions about global events.

**Setup.** Let $\mathcal{X}$ be the set of open-ended forecasting questions; and $\mathcal{Y}$ the set of short textual answers. We provide a language model $\pi_\theta$ a question $x \in \mathcal{X}$, for which we already know the ground-truth outcome $y^\star$ as it has resolved in the real-world. We ask the model to respond with its best guess answer $y$, and the probability $q$ the model assigns to that being the true outcome.

**Measuring Accuracy.** We measure accuracy by checking if the model's attempted answer $y$ matches with the ground truth outcome $y^\star$, using another language model to test for semantic equivalence (for example "Geoffrey Hinton" = "Geoffrey Everest Hinton") consistent with recent frontier benchmarks (Wei et al., 2024; Phan et al., 2025). For evaluations, we use Llama-4-Scout (Meta AI, 2025), as in a recent study (Chandak et al., 2025), it aligns with human judgments when matching answers at an inter-human level. For training we use Qwen3-4B in non-thinking mode, as it achieves high alignment levels for its size (Chandak et al., 2025). We find the two models agree on $\sim 97\%$ responses graded, and human validation ensures they are accurate in $\geq 95\%$ cases, c.f. Appendix D.

**Measuring Calibration.** We adapt the multi-class Brier scoring rule (Mucsányi et al., 2023) for free-form response as follows (details in Appendix A):

$$S'(q, y, y^*) = \begin{cases} 1 - (q-1)^2, & \text{if } y \equiv y^* \\ -q^2, & \text{if } y \neq y^* \end{cases}$$

This score has a natural interpretation: predicting an event with a probability $q = 0$ returns a baseline score of 0 regardless of the guess $y$ of the event. Correct predictions receive positive scores while incorrect predictions negative. For brevity, we call $S'(q, y, y^*)$ *Brier score* throughout this paper. Our Brier score is equivalent to the reward metric used by Damani et al. (2025). They show this is a proper scoring rule, incentivizing both high accuracy and truthful reporting of probability on the answer that seems most likely. For completeness, we discuss this further in Appendix A.

**Training Algorithm: GRPO (Shao et al., 2024).** We train LLMs using outcome-based reinforcement learning on our dataset. For each prompt $x$, we draw $K$ completions $\{(y_i, p_i)\}_{i=1}^K \sim \pi_\theta(\cdot \mid x)$ and compute rewards $r_i = R(y_i, p_i; y^\star)$. However, following prior work (Damani et al., 2025; Turtel et al., 2025b), we *remove* the per-group standard-deviation division during the advantage computation as it stabilizes updates in settings like ours where reward variance can sometimes be too small.

**Initial Policy: Qwen3 Thinking (Yang et al., 2025).** We start with the 4B and 8B thinking models. For Qwen3 models, no official knowledge–cutoff date is reported. When queried directly, the models return inconsistent cutoff dates (most often *October 2023* or *June 2024*), often treating questions about 2024 as being in the future. Since the model weights were released and frozen in April 2025, we train up to this date, and use the period between May to August 2025 for testing.

## 4 GENERATING OPEN-ENDED FORECASTING QUESTIONS FROM NEWS

We now discuss our methodology to convert daily news articles into forecasting questions for language models. Any fixed forecasting dataset loses value as newer base models get adopted which have training cutoffs after the dataset was created. Thus, we first describe the general methodology which can be repeated in the future, and then describe the specific instantiations we used to create our training data `OpenForesight` which has questions until March 2025. We conclude by demonstrating improvements in training enabled by our data filtering steps.

### 4.1 METHODOLOGY FOR GENERATING FORECASTING QUESTIONS

We generate short-answer, open-ended forecasting questions from individual news articles as illustrated in Figure 2. We describe each step in detail below:

**Sourcing Event Information.** News outlets are an established global engine for reporting salient events as they occur. Unfortunately, Paleka et al. (2025a) show that sourcing them via online search engines is unreliable. While search engines provide date cutoffs, future information can even leak through search engine ranking, and updates to articles after the publish date. This compromises the reliablity of backtests, and leaks future information in training, which can hurt Deep Learning

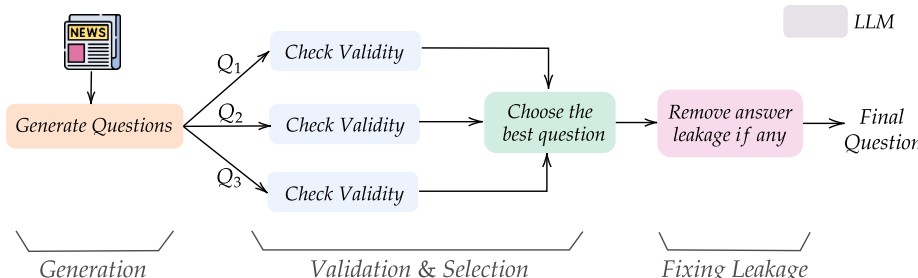

Figure 2: **Our question generation methodology.** We use DeepSeek-v3 to generate multiple forecasting questions per news article. Then, we use a different model, Llama-4-Maverick, to check if questions follow all guidelines, choose the best question, and remove any hints revealing the answer.

models which easily overfit to spurious correlations. Fortunately, the CommonCrawl News (CCNews) Corpus (Nagel, 2016) provides static monthly snapshots of global news with accurate dates. This makes it free and easy to obtain news articles for creating forecasting questions.

**Generating questions from documents.** Based on each news article, we ask a language model to generate up to three diverse forward-looking forecasting samples. Each sample consists of: (i) a concise question about an event with an explicit deadline (e.g., "by *Month, Year*"); (ii) brief background that provides context, or defines uncommon terms; (iii) resolution criteria that fixes a source of truth and the expected answer format; (iv) The unique answer, drawn verbatim from the article, usually short (1–3 words), non-numeric (usually a name or location); and (v) Source article link for reference, obtained from article metadata. We show an example in Appendix C.

**Filtering questions.** For each question, we use another LLM to verify the following properties: (i) the question-answer pair is fully based on information in the source article (ii) the question is in future tense and (iii) the answer is definite, unambiguous, and resolvable by the publication date. We mark a question as valid only if it passes these checks. If multiple questions from a single article remain, we use another model to select the best one to further improve data quality and diversity. We ask it to favor questions with clear, unique answers and high relevance.

**Editing to fix leakage.** At this stage, we find that even the filtered samples sometimes leak information about the answer. This can create shortcuts during training. To fix this, we do a final editing stage where we use an LLM to scan the title, background, and resolution criteria to check if they reveal the answer. When it finds leakage, we ask it to rewrite only the offending spans, replacing specifics with generic placeholders. Finally, we re-scan using exact string matching any remaining mentions of the answer, and discard those question-answer pairs.

Overall, this pipeline can continually ingest news articles and generate high-quality open-ended forecasting questions for training. We use the same methodology but *different news sources* to create a validation and test set, to ensure our forecasting systems learn generalizable forecasting skills.

## 4.2 OPENFORESIGHT: AN OPEN, LARGE-SCALE FORECASTING TRAINING DATASET

We now describe the specific composition of our training dataset.

**Generating questions.** One practical issue we face is that many top news sources, such as The Reuters and Associated Press (AP), have disallowed scraping even for CommonCrawl, due to the rise of commercial use in language model training (Grynbaum & Mac, 2023; Longpre et al., 2025). Still, we are able to collect articles from popular outlets spanning diverse geographies and topics. Particularly, for our training set, we start with $\sim 248,000$ deduplicated English-language articles between June 2023 to April 2025 from *Forbes*, *CNN*, *Hindustan Times*, *Deutsche Welle*, and *Irish Times*. The distribution is described in Table 3. From these, we generate three forecasting-style questions per article using DeepSeek v3, yielding $\sim 745,000$ question–answer candidates.

**Filtering questions.** For all further data filtering, we use a different model, Llama-4-Maverick to prevent leniency caused by LLM self-preference (Xu et al., 2024). Table 1 contains a breakdown of questions remaining after each filtering stage.

60% of question-answer candidates are marked invalid—most commonly because the article does not unambiguously resolve the question to the given answer. At this stage, zero questions remain from 40% articles, and 21% articles yield exactly one valid question, which we keep as is. For the 39% with multiple valid questions, we ask the model to pick the best one. Finally, to avoid vague or numeric answers, we only keep questions with specific types, listed in Table 4.

| Stage | Number (%Total) |
|---|---|
| Source Articles | 248,321 |
| Question Generation | 744,963 (100%) |
| Validation | 295,274 (40%) |
| Best Question Selection | 157,260 (21%) |
| Fixing Leakage | 150,500 (20%) |
| Answer Type Filtering | 62,279 (8%) |
| Final Set | 62,279 (8%) |

Table 1: Number of questions after each filtering stage.

**Editing to fix leakage.** Despite explicit prompts to avoid it, over $40\%$ of selected questions directly contain the answer string. In the step where we use Llama-4-Maverick to rewrite or reject questions with leakage, we are able to remove $\sim 90\%$ of such cases. We then apply a string matching filter to remove the remaining questions with such direct leakage.

**Ablation: Effect of filtering.** To measure the effect of our filtering steps, we train Qwen3-8B using RL with identical hyperparameters on three data variants. The first consists of 10,000 samples sourced from Forbes and included in `OpenForesight`. The second consists of all 30,000 questions generated originally from their respective articles, without any filtering. The third also consists of 30000 samples on which we perform the question editing step to remove leakage.

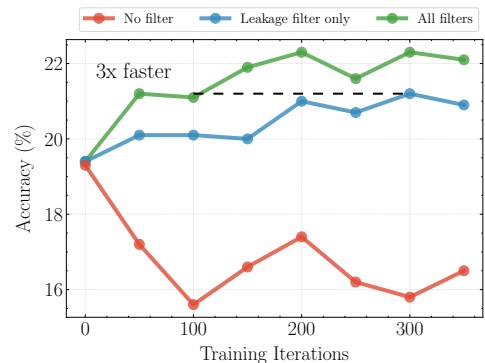

**Result 1: Filtering Improves Performance and Learning Efficiency**. We observe the effect of different stages of filtering in Figure 3. First, we observe the drastic impact of leakage in training. Training without leakage removal (red line) worsens the model, perhaps due to shortcut learning. After the leakage removal steps, training improves the model (blue line). Yet, using all filtering stages (green line) leads to both higher accuracy and Brier score, in 3x less data and half the iterations. This result demonstrates the importance of data quality for training LLMs for forecasting with RL.

Figure 3: **Benefits of our filtering recipe.** Without leakage removal (red), we model does not improve at forecast, possibly learning shortcuts. Without filtering (blue), we find that achieving the same performance requires 3x more compute and data. Applying all filtering steps (green) leads to higher final performance across both metrics.

**Final training dataset.** Across stages, we remove $\sim 90\%$ of questions, yielding a high-precision set of 62K question-answer pairs, each drawn from a unique article. Evaluating Qwen3-32B on these pairs *with the respective source article* yields 95% accuracy, confirming dataset validity. We will release this training dataset, `OpenForesight`, to promote research on open-ended forecasting.

In Appendix B.1, we also ablate the effect of training on binary-only, free-form-only, and combined binary and free-form data for `Qwen3-8B`. We find that free-form data is crucial for improving open-ended forecasting but training solely on freeform data does not improve performance on binary Metaculus questions. Training with both kind of questions achieves the best trade-off.

## 5 PREDICTION SYSTEM

We now present intermediate results that guided the design decisions for our prediction system. This includes designing a retrieval system to obtain relevant documents for each question, an SFT warm up stage, and designing the reward for RL training. We did not measure performance on the held-out test set throughout this process. Instead, we used the same data curation recipe described in Section 4 to generate a validation set of 207 questions generated using The Guardian articles from July 2025.

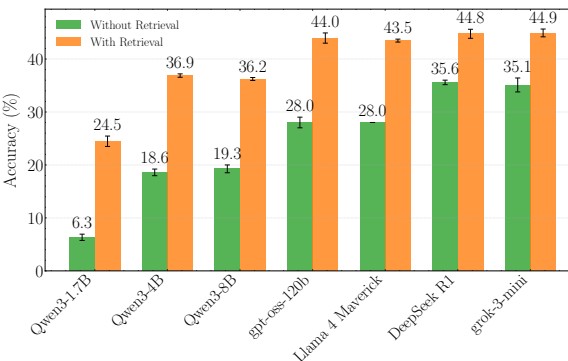 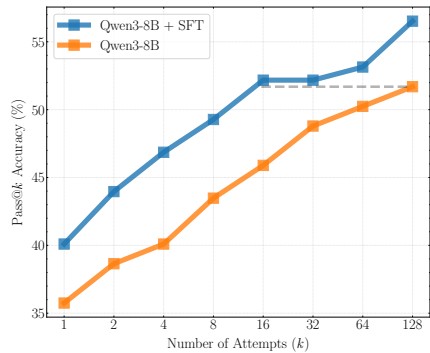

Figure 4: **Retrieval improves accuracy significantly with models ordered by their size.** We use the specialized Qwen3 8B embedding model for this. We take a cautious approach, retrieving relevant articles only until a month before the resolution date. We embed up to 5 articles in the prompt of the model.

Figure 5: **SFT improves both pass@k and pass@1.** After SFT distillation on Grok-3-mini traces, we find that pass@16 of the SFT model surpasses even pass@128 of the original 8B model on our validation set.

**Retrieval.** Like prior work (Zou et al., 2022; Halawi et al., 2024), we retrieve relevant recent documents to assist the model's forecast. This gives it access to information, like new evidence, or competing viewpoints to weigh, that could affect the answer known after its training cutoff. To prevent leakage issues (Paleka et al., 2025a), we use our offline CCNews corpus of articles, and only provide retrieved articles up to *one month* before the question's resolution date. Our overall pool consists of 1 million articles across 60 different sources. We de-duplicate the articles and split each into fixed-size chunks (512 tokens) and embed each chunk with the Qwen3-embedding 8B model.

**Result 2: Our retrieval significantly improves accuracy.** As shown in Figure 4, our retrieved articles improve accuracy by 9 to 18% across model families and sizes. In Appendix Figure 11, we vary the number of retrieved articles for Grok-3-Mini and find that the improvement plateaus after 5 articles. Thus, we use 5 articles for training and evaluation, unless specified otherwise.

**Supervised Finetuning (SFT).** Even though we start from the RL trained Qwen3 thinking models, they are far behind proprietary models as shown in Figure 4. Several frontier model training reports (Guo et al., 2025) mention using an SFT stage as a warm start before RL. We choose Grok-3-Mini to generate forecasting reasoning traces for SFT, as it has high performance, low cost, and provides the full reasoning trace through the API. Specifically, we construct a dataset of 10,000 questions from *The Guardian* dated January–March 2025, beyond Grok-3-mini's reported knowledge cutoff of June 2024. Obtaining Grok-3-Mini's reasoning traces on this data costed 15 dollars. To test the usefulness of SFT for eventual GRPO, we compute pass@k accuracy (Wu et al., 2025), which measures the fraction of samples where the model gets at least one attempt out of $k$ correct.

**Result 3: SFT improves pass@k performance of the model.** Figure 5 shows pre and post-SFT pass@k results. We observe SFT consistently improves both pass@1 and pass@k accuracy, ensuring little diversity collapse. We thus decide to use SFT to distill Grok-3-mini reasoning traces into our Qwen3-8B model before further RL training.

**Reward Design.** For training with RL, we investigate three reward functions:

1. **Baseline.** Only Accuracy: $R = \mathbb{1}_{y \equiv y^*}$. Binary success rewards are commonly used in literature on LLM RL with verifiable rewards (Guo et al., 2025).

2. **Damani et al. (2025).** Only Brier score: $R = S'(q, y, y^*) = -q^2 + \mathbb{1}_{y \equiv y^*} \cdot 2q$. From Section 3, this incentivizes both correct predictions and calibrated confidence estimates.

3. **Ours.** Accuracy + Brier score: $R = \mathbb{1}_{y \equiv y^*} + S'(q, y, y^*)$. We hypothesise optimizing the Brier score alone hurts exploration as when the model assigns a low confidence to its guess, the correctness of the prediction has a small impact on the Brier score. To fix this, we propose adding the accuracy term as well. In this case, even on hard questions which merit low confidence, if a model makes a correct prediction, it would get a significant boost in reward.

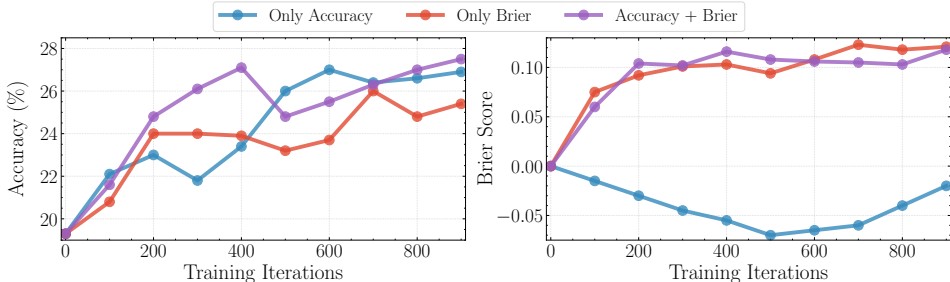

Figure 6: **Accuracy + Brier score reward performs the best.** Accuracy alone leads to poor calibration. While brier incentivizes both correct predictions and calibration, the extra boost from success incentivizes the model to try its best guess with low probability on hard questions.

**Result 4: Accuracy + Brier improves RL, incentivizing exploration.** Figure 6 shows the validation set results of training with all three reward functions on the full `OpenForesight` dataset, without retrieval. We observe that optimizing accuracy alone leads to negative brier scores, worse than a constant (0) baseline. In contrast, the optimizing the Brier score alone also improves the accuracy. Our proposed reward, accuracy + Brier, performs the best. It improves accuracy beyond the brier alone while maintaining obtaining equal brier score on the validation set. Analyzing output distributions, we find that the brier-only trained model predicts "Unknown" with near-0 confidence in $\sim 40\%$ of samples, due to low reward for correct yet low-confidence guesses, which hurts exploration. In contrast, our proposed reward yields "Unknown" in only $\sim 4\%$ of samples, making low-confidence guesses on hard cases—improving both accuracy and training efficiency.

**Training the final forecasting system.** Based on the above design decisions guided by validation set performance, we now describe our final training methodology: We use the Qwen3-8B embedding model to retrieve the 5 most relevant chunks from news articles until a month before each question's resolution date. We use SFT to distill on 10,000 Grok-3-mini generated reasoning traces on questions between from January to March 2025. We then train this checkpoint with GRPO on `OpenForesight` which has 60,000 samples and also include 2000 binary resolved questions from Metaculus (from 2024), both with retrieval (top-5 article chunks added in the prompt). For the reward, we use our Accuracy + Brier score for free-form questions and only brier score for binary questions.

# 6 FINAL RESULTS

We now present evaluations of our models, `OpenForecaster` 4B and 8B. To avoid making decisions based on future information, we evaluate on test sets that were not observed until the end.

**Evaluation Datasets.** Typically, existing LLM forecasting benchmarks do not provide open-ended questions, and suffer from distributional biases highlighted in Paleka et al. (2025a). Many others (Wang et al., 2025) only have questions that are no longer "in the future" for our models. Among recent ones, we try using the resolved subset of non-numeric questions from parallel work, the FutureX benchmark (Zeng et al., 2025). However, we find both small and frontier models have very similar performance as shown in Appendix Figure 7b, with large standard deviations as there are only 86 usable questions. So we evaluate our trained models on three more types of datasets.

First, we use our data curation recipe to create a test set of 1,000 questions between May to August 2025. To ensure high-quality evaluation, we use `o4-mini`, a much more capable model than DeepSeek-v3, to generate the seed questions. Crucially, we also use five distinct, diverse news sources: Al Jazeera English (global news, based out of Qatar), Time (global news, based out of USA) The Independent (UK focused), Fox News (USA focused), NDTV (India focused), with 200 questions generated from each. We deliberately use distinct sources from the training set to ensure that our model is learning generalizable forecasting skills, and not source distribution specific biases. The choice of sources was made under the constraint of many established news sources disallowing crawling of their articles starting 2025. Second, for evaluating on long-term predictions, we measure consistency using the dataset and methodology proposed by Paleka et al. (2025a) which are shown to strongly correlate with forecasting performance. Finally, to measure whether our forecasting training generalizes to calibration on standard benchmarks of LLM capabilities, we evaluate, without

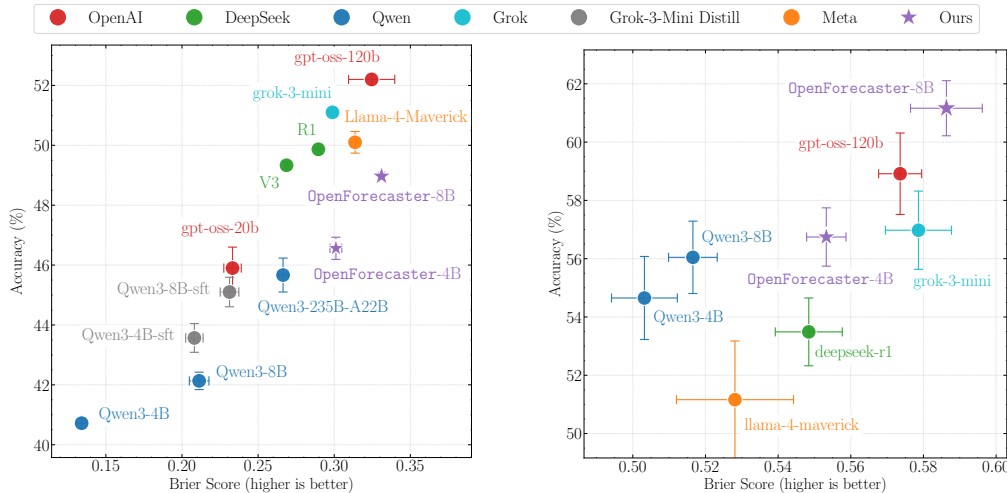

(a) Performance on our open-ended forecasting test set.    (b) Performance on the FutureX benchmark.

Figure 7: **Our forecasting training jointly improves accuracy and calibration** both on open-ended questions in our test set, and the external FutureX benchmark, making `OpenForecaster` 8B competitive with much larger models with cutoffs before May 2025.

retrieval, on SimpleQA (Wei et al., 2024), a challenging factuality benchmark, and MMLU-Pro and GPQA-Diamond which are popular cross-domain reasoning benchmarks.

**Result 5: Our training significantly improves forecasts.**    Figure 7a shows performance of models on our held-out test set of open-ended forecasting questions. On the Brier score (X axis), the primary metric recommended for forecasting (Tetlock & Gardner, 2016) as it measures both accuracy and calibration, `OpenForecaster` 8B outperforms the much larger proprietary models we tested, and the 4B model matches them. Our improvements are not merely from calibration, the predictions also become more accurate (Y axis), though they are a bit behind the larger models. Both the SFT (grey markers labelled with "-sft"), and RL (purple markers labelled with "Open-Forecaster") improve both the brier score and accuracy of our forecasting system, with the latter leading to larger absolute gains in performance. We also show model accuracy by month in Figure 13.    Further, on the benchmark proposed by Paleka et al. (2025b) consisting of binary questions resolving in 2028, `OpenForecaster` 8B makes more consistent long-term predictions, 44% more on arbitrage metrics, and 19% more on frequentist metrics, across all ten consistency checks. See Appendix B.3 for detailed results. Our data can also be used to improve models from other families like Llama and Gemma family on `OpenForesight` as we show in Appendix B.2.

We saw a particularly large (+25% accuracy) improvement for Llama 3.1 8B Instruct, even surpassing the much larger Qwen3-235-A22B.

**Improvements on External Forecasting Benchmarks.**    We also validate our models on resolved questions from the existing FutureX (Zeng et al., 2025) dataset[1]. We filter to English and non-numeric questions, which leaves only 86 binary or multiple choice samples. In Figure 7b, we plot the performance of the models. Training on `OpenForesight` leads to large performance improvements, making our 8B model outperform GPT-OSS-120B.

Finally, we also simulate live-testing of our frozen checkpoints from September on Metac-

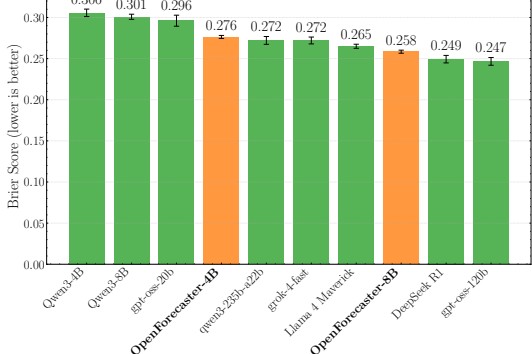

Figure 8: Brier score of models on Metaculus questions from October 01 to November 18 2025.

[1] https://huggingface.co/datasets/futurex-ai/Futurex-Past

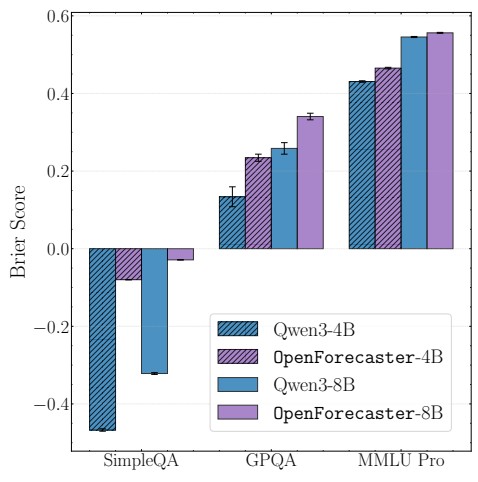 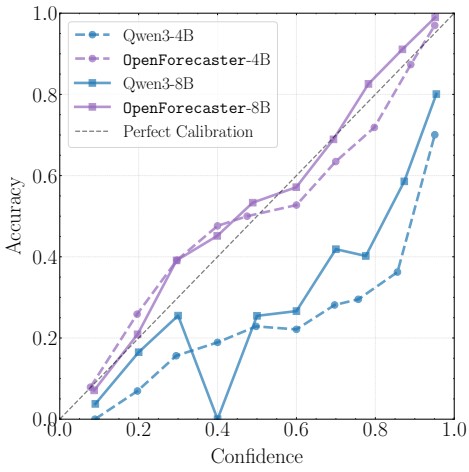

(a) Brier score on general benchmarks.

(b) Calibration curves on our test set.

Figure 9: Calibration of the models improve significantly after training on `OpenForesight` both on (a) OOD benchmarks and (b) on our test set.

ulus from October 01 to November 18 2025. In
this time window, we obtain 160 questions in from the Metaculus API. We filter questions which were related to stock price prediction, or meta-prediction about how Metaculus questions would resolve, as this requires access to very recent information not supported by our local offline news retrieval which is updated on a monthly cadence. This left us with 69 samples. As shown in Figure 8, all models we tested performed equal or worse than a random baseline brier score of 0.25, perhaps due to the questions still requiring more recent retrieval. Still, our training does improve the brier score even on this live test, making `OpenForecaster` 8B surpass Qwen3-235-A22B and GPT-OSS-20B.

**Result 6: Calibration training for forecasting generalizes to factuality.** Figure 9a shows downstream improvements in calibration across SimpleQA, GPQA-Diamond and MMLU-Pro. This calibration can then be used to reduce hallucinations, for example abstaining on questions the model is not confident about, using a simple rule like `if probability < 0.1, replace prediction with "I do not know"`

**Summary.** On both the 4B and 8B scale, GRPO training with our proposed reward for forecasting delivers large gains in both Brier score and accuracy, making small specialized models competitive with large general ones like DeepSeek R1 and gpt-oss-120B. Improvements in calibration generalize to a challenging downstream factuality dataset.

## 7 CONCLUSION

In this paper, we take the first step towards *scalable training* for *open-ended forecasting*. The results are promising, we significantly improve both accuracy and Brier score, matching a much larger 670B model by finetuning an 8B model. A few limitations remain. For example, we only use news to create forecasting questions, which leads to a distributional bias. The news also reports some events late, such as scientific breakthroughs, and this can make such questions easier to "predict" than others by their resolution date in our dataset. This should not affect relative performance comparisons between models though. We also do not consider generative, long-form forecasts, as it is unclear how to grade these. Overall, open-ended forecasting, being a challenging and highly valuable task, offers exciting directions to pursue across research communities. A strong forecaster needs to reason about uncertainty, efficiently seek new information, and make optimal Bayesian updates to its world model, long-standing challenges in the quest for general intelligence. Scaling up end-to-end training of language model based forecasting systems may lead to emergent improvements in such capabilities. By open-sourcing all our artefacts, we hope to spark more research on this important direction.

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

# Appendix

CONTENTS

## A   ADAPTING BRIER SCORE TO FREE-FORM RESPONSES

We evaluate probabilistic predictions using the Brier score (Mucsányi et al., 2023). For a $K$-class outcome space $\mathcal{Y}$ with reported distribution $q$ and true class $y^*$, the (multi-class) Brier score is

$$S(q, k) = -\sum_{y \in \mathcal{Y}} (q_y - k_y)^2 = -(q_{y^*} - 1)^2 - \sum_{y \neq y^*} q_y^2,$$

where $k$ is one-hot with $k_{y^*} = 1$. In our open-ended setting, $\mathcal{Y}$ is not predefined but rather its instances are provided by the forecaster. For simplicity, we elicit only a single guess $y$ with confidence $q \in [0, 1]$. Applying the multi-class brier scoring rule in such a case induces a simplified score:

$$S(q, y, y^*) = \begin{cases} -(q-1)^2 - 0 = -1 + 2q - q^2, & \text{if } y \equiv y^*, \\ -(0-1)^2 - q^2 = -1 - q^2, & \text{if } y \neq y^*. \end{cases}$$

Dropping the constant $-1$ yields

$$S'(q, y, y^*) = \begin{cases} 1 - (q-1)^2, & \text{if } y \equiv y^*, \\ -q^2, & \text{if } y \neq y^*, \end{cases}$$

which shifts the range from $[-2, 0]$ to $[-1, 1]$ while providing a more natural interpretation: predicting $q = 0$ gives a baseline 0 regardless of $y$; correct answers receive positive scores, incorrect answers negative scores; and magnitude scales quadratically with confidence. We report $S'$ as the *Brier score* in this paper.

Recent work by Damani et al. (2025) shows that this metric is a proper scoring rule, incentivizing both high accuracy and truthful reporting of probability on the answer that seems most likely. However, note that what we call as brier score here is distinct from the brier score considered by Damani et al. (2025). Their brier score is the one traditionally used for evaluating binary outcomes while ours is for free-form responses. Yet, our brier score is same as the training reward considered by them.

## B   ADDITIONAL RESULTS

### B.1   ABLATION: COMPARISON TO PREDICTION MARKET BINARY DATA

We ablate supervision type with Qwen3-8B using three size-matched settings (Figure 10). For *binary-only*, we curate **20K** resolved markets from Manifold, volume-filtered to ensure engagement; because many markets resolve slowly, this set spans the past five years. For *free-form only*, we use **20K** pipeline-generated, usable questions from Forbes articles. For the *binary+free-form mix*, we take **10K** Manifold + **10K** Forbes questions to keep total examples constant. The goal is to isolate which *learning signal*—binary resolution vs. open-ended outcome specification—most effectively trains calibrated forecasters under identical compute and token budgets.

On the free-form test set (Fig. 10 Left), post-RL performance improves most with *free-form only* supervision (Accuracy $19.3\% \to 22.4\%$; Free-form Brier $-0.009 \to 0.086$). Mixing binary and free-form also helps (Brier 0.065), whereas *binary-only* yields minimal gains on free-form evaluation (Brier 0.004). On Metaculus (binary) (Fig. 10 Right), both *binary-only* and the *mixed* setting improve accuracy and Brier, with the *binary+free-form* mix offering the best overall trade-off across testing formats. Our gains by training on binary-only format are consistent with prior work by Turtel et al. (2025b;a). However, we do not arrive at a single unanimous recipe: free-form data is essential for open-ended forecasting, while combining formats appears Pareto-optimal across binary and free-form evaluations. Practically, it seems training on a *mixture* of question styles provides the most robust gains across tasks.

### B.2   RESULTS ON FREE-FORM FORECASTING

In Figure 11 we observe that while the first few article chunks that are retrieved to large improvements, at around five articles improvements plateau, both on the Qwen3-8B and Grok-3-mini models used during distillation. Thus, unless otherwise specified, we use 5 articles for all evaluations and training in this work.

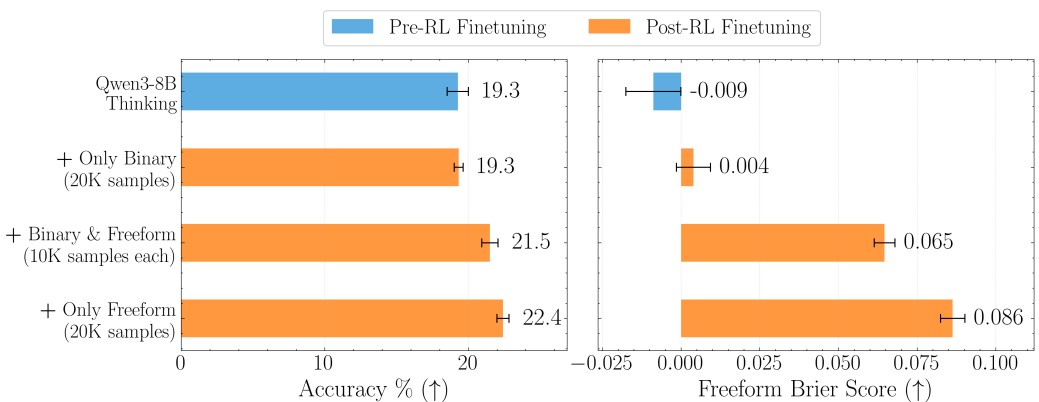

(a) Performance on our **Validation Set** composed of question from TheGuardian new source from July 2025.

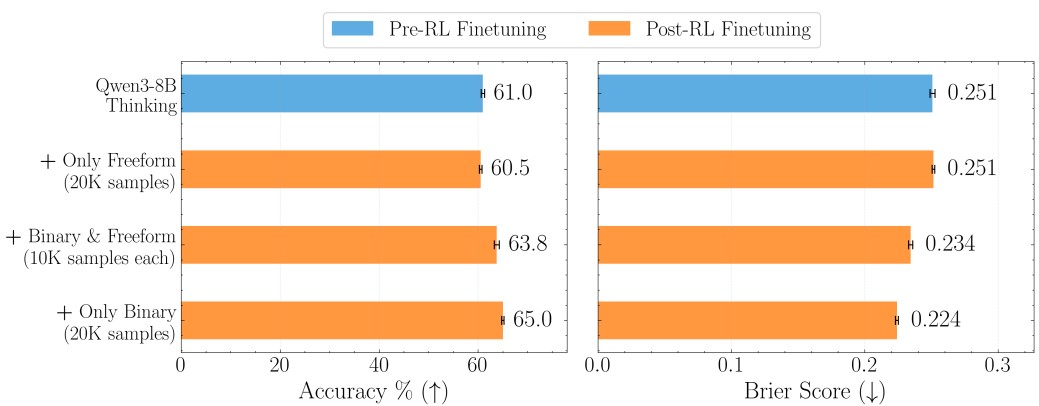

(b) Performance on **Metaculus binary** questions resolved in May–July 2025.

Figure 10: **Performance of different data ablations.** We evaluate performance after training on 3 different supervision signals: (i) only binary data (20K samples), (ii) only freeform data (20K samples), and (iii) both binary and freeform data (10K samples each) for data-matched comparison. (a) Accuracy and freeform Brier score of the initial and post-RL model on our Validation Set from July 2025. (b) Accuracy and binary Brier score of initial and post-RL model on volume-filtered binary questions resolved between May to July 2025 on Metaculus. *We find training on binary questions hurts performance on open-ended forecasting, but is necessary to retain performance on binary prediction market questions.*

**Results on Validation Set.** We report results on our validation set based on TheGuardian (207 questions) for our final model, showing significant improvements from training, and that it is competitive with much larger models, consistent with Figure 7a.

**Results over time.** As our test set is derived from articles from May to August 2025, so we split the questions by resolution date to get monthly performance of the models. Breaking down by month, our test has 270 questions resolving in May, 265 in June, 193 resolving in July and 137 resolving in August. Our hypothesis is that as we go further into the future, forecasting should become more difficult leading to lower performance. In Figure 13 and Figure 14,

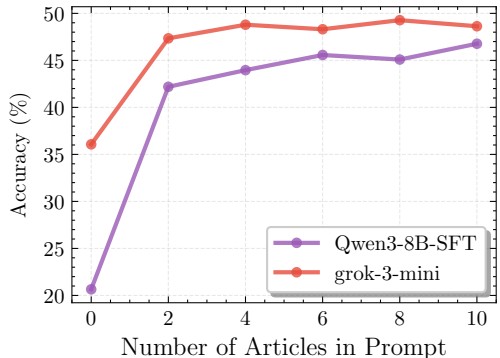

Figure 11: **Improvements from retrieval plateau at** $\sim 5$ **chunks.** We show the accuracy of both Grok-3-mini, the teacher model we use for the warm-up phase, and the Qwen3 8B model after distillation from it.

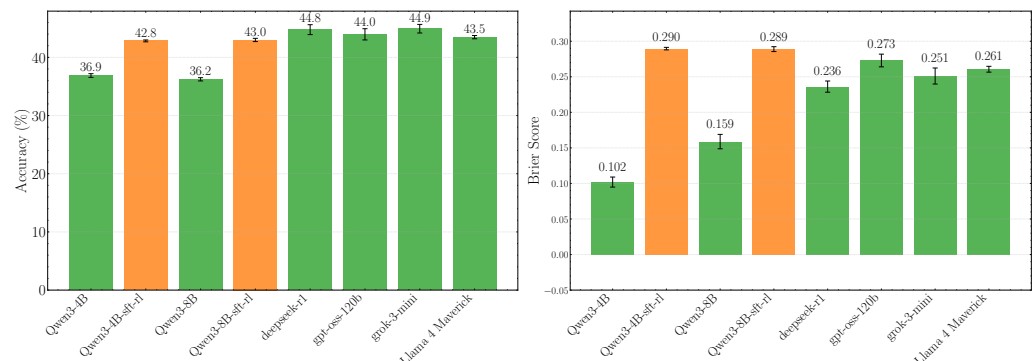

Figure 12: Performance of the models on our validation set.

we find that the accuracy and brier score of the
models indeed drops gradually month-by-month
consistent with our hypothesis. We also find that
our trained models are consistently better than the original versions and also better than all other
models in Brier score.

**Improvement on non-Qwen models.**   Our training data `OpenForesight`can be used to improve
models across different families. In Figure 15 we show improvements for Llama-3.1-8B-Instruct,
Llama-3.2-3B-Instruct and Gemma-3-4B-Instruct. We see particularly large improvements in both
accuracy and brier score for Llama due to both: poor initial performance, but also surprising
amenability to RL training with our data as the final performance exceeds GPT OSS 20b. We also
provide a qualitative analysis of the change in performance of LLama-3.1-8B in Appendix F.

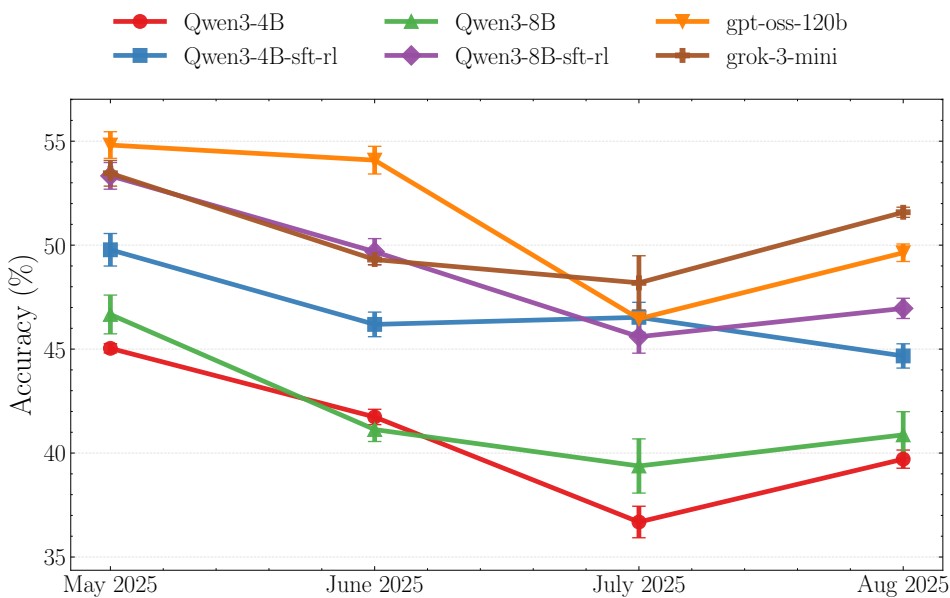

Figure 13: Monthly accuracy of the models on our test set. Across models, we observe consistent trends that indicate questions in our test set from July are significantly harder than others.

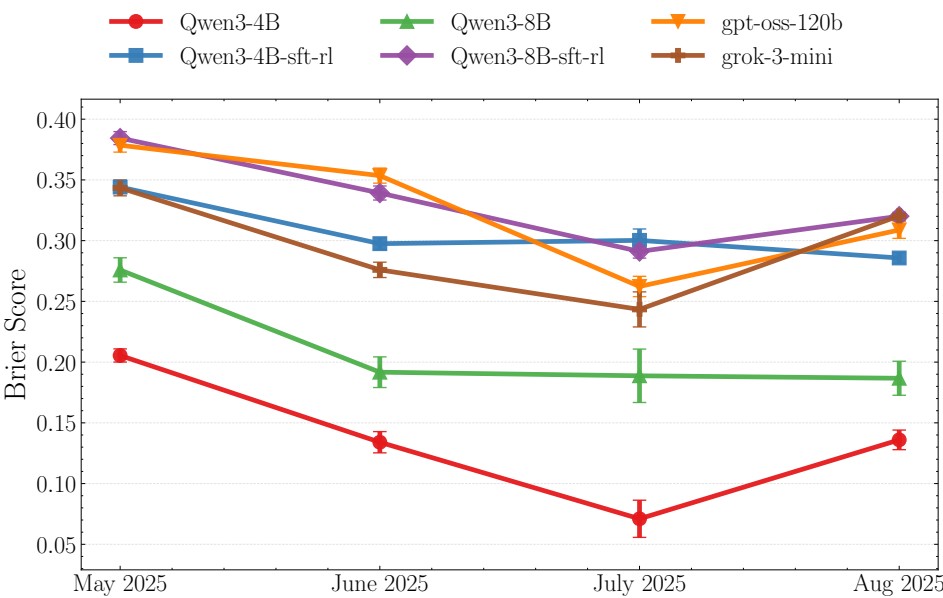

Figure 14: Monthly brier score of the models

## B.3 CONSISTENCY EVALUATION

Paleka et al. (2025b) release a dataset of long-term forecasting questinos set to resolve up to 2028, showing language models exhibit inconsistencies in their probabilistic predictions. To evaluate consistency, they propose ten consistency checks measuring both arbitrage and frequentist violations.

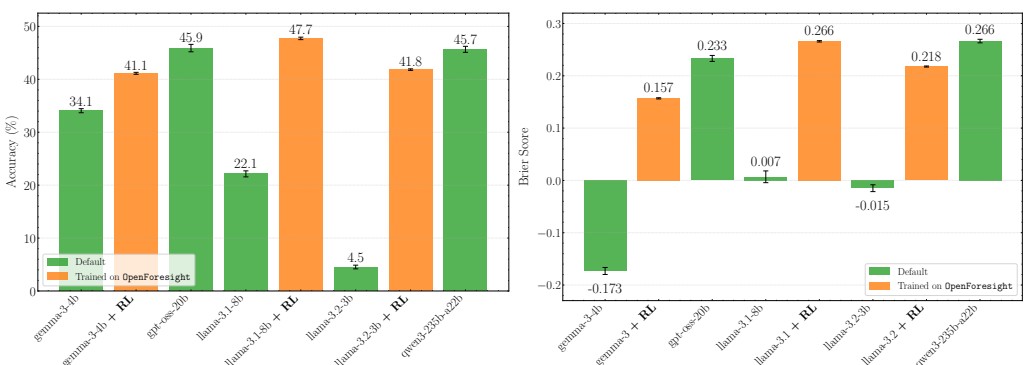

Figure 15: Performance of models from Llama and Gemma family our test set.

We evaluate Qwen3-8B and our trained model on the dataset created by Paleka et al. (2025b). We measure performance of the models on all consistency check tuples proposed by them. Table 2 compares the baseline Qwen3-8B with our RL-trained model. The results demonstrate substantial improvements across most consistency checks, with particularly strong gains in Boolean logic operations (AND: 78% reduction, OR: 64% reduction) and paraphrase consistency (50% reduction). Overall, our training achieves a 43.5% reduction in arbitrage violations and 19.2% reduction in frequentist violations, indicating more consistent long-term predictions.

Table 2: **Improvement in consistency checks before and after RL training**. We report average violation scores and relative improvements (negative percentages indicate improvements). The RL-trained model shows substantial improvements in logical consistency across most reasoning tasks.

| Check | Arbitrage | | | Frequentist | | |
|---|---|---|---|---|---|---|
| | Qwen3-8B | OpenForecaster-8B | Δ | Qwen3-8B | OpenForecaster-8B | Δ |
| NEGATION | 0.043 | 0.029 | -32% | 0.198 | 0.177 | -11% |
| PARAPHRASE | 0.030 | 0.015 | -50% | 0.157 | 0.114 | -27% |
| CONSEQUENCE | 0.010 | 0.003 | -66% | 0.048 | 0.033 | -31% |
| ANDOR | 0.033 | 0.019 | -43% | 0.205 | 0.148 | -28% |
| AND | 0.016 | 0.004 | -78% | 0.063 | 0.026 | -59% |
| OR | 0.022 | 0.008 | -64% | 0.094 | 0.061 | -35% |
| BUT | 0.040 | 0.021 | -47% | 0.234 | 0.193 | -17% |
| COND | 0.039 | 0.030 | -23% | 0.227 | 0.220 | -3% |
| CONDCOND | 0.036 | 0.032 | -13% | 0.256 | 0.255 | -0% |
| EXPEVIDENCE | 0.041 | 0.015 | -64% | 0.240 | 0.166 | -31% |
| **Aggregated** | **0.031** | **0.017** | **-44%** | **0.172** | **0.139** | **-19%** |

## C DATASET DETAILS

---

**Sample Generated Forecasting Question**

**Question.** Who will be confirmed as the new prime minister of Ukraine by 17 July 2025?

**Background.** Ukraine's parliament is scheduled to vote to appoint a new prime minister.

**Resolution Criteria.**

- **Source of Truth**: Official announcement from the Verkhovna Rada (Ukraine's parliament) confirming the appointment, via parliamentary records or government press release.

- **Resolution Date**: 17 July 2025, the date on which the parliamentary vote occurs and results are published.

- **Accepted Answer Format**: Full name of the individual exactly as given in the parliamentary announcement.

---

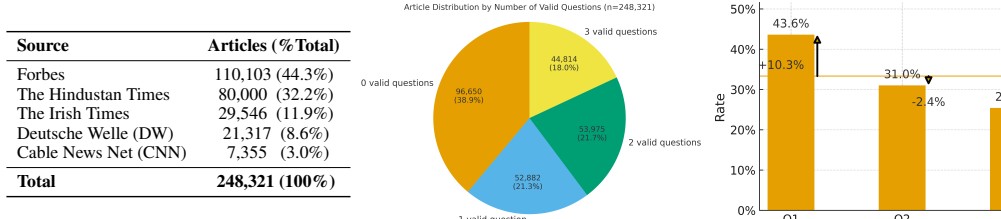

| Source | Articles (%Total) |
|---|---|
| Forbes | 110,103 (44.3%) |
| The Hindustan Times | 80,000 (32.2%) |
| The Irish Times | 29,546 (11.9%) |
| Deutsche Welle (DW) | 21,317 (8.6%) |
| Cable News Net (CNN) | 7,355 (3.0%) |
| **Total** | **248,321 (100%)** |

Table 3: **Data Distribution of `OpenForesight`.** (Left) We show the breakdown of source documents by news outlet. (Right) We show the number of questions generated, and the proportion of the first, second and third generate question being picked as the final "best question".

| | Name(s) | Location | Country | Title | Team name | Color | Organization | Currency | Brand name | Month |
|---|---|---|---|---|---|---|---|---|---|---|
| **Count** | 32,213 | 14,337 | 2,579 | 2,479 | 1,445 | 1,047 | 1,030 | 877 | 779 | 730 |
| **Share** | 44.8% | 20.0% | 3.6% | 3.5% | 2.0% | 1.5% | 1.4% | 1.2% | 1.1% | 1.0% |

Table 4: Top ten answer types of the questions in our curated dataset. These ten categories cover **80.1%** of our training dataset.

---

**Answer Type.** String (Name)

**Ground-Truth Answer.** Yulia Svyrydenko

**Source.** The Guardian (live blog): Ukraine live updates — 17 July 2025

---

## D    QUALITATIVE ANALYSIS OF FINAL ANSWERS

We manually annotated responses to 207 questions by both the initial Qwen3-8B thinking model and the trained `OpenForecaster` 8B on the Guardian validation set. Using this set, we found that the agreement between the two models used for grading, Llama 4 Scout and Qwen3 4B is $\sim 97\%$, and we agree with their grading in over $\sim 95\%$ cases. This confirms the reliability of automatic answer matching based evaluation.

In Table 6, we analyze the domains (by news section) in which our trained model improves. We find significant improvements in the World, Australian, and US news sections, with no significant change for sports. This entails our model may not yet perform too well on sports-heavy prediction markets like Kalshi.

In Table 7, we analyze change in performance by question type, finding significant improvements on questions of the form "what", "which", and "who", while a slight regression in performance on location questions ("where").

Below, we present qualitative examples where our training improves and worsens predictions compared to the original model.

QUALITATIVE EXAMPLES (IMPROVED; FIRST SAMPLE)

- **Q:** Who will be wearing the yellow jersey in the general classification at the end of stage eight of the 2025 Tour de France?
  **Truth:** Tadej Pogacar
  *Before:* Jonas Vingegaard (p=0.10)
  *After:* Tadej Pogacar (p=0.60)

- **Q:** Who will withhold a resolution from the U.S. House floor to force a vote on releasing the Epstein documents by July 25, 2025?
  **Truth:** Mike Johnson
  *Before:* Pam Bondi (p=0.30)
  *After:* Mike Johnson (p=0.60)

| Question | Background | Resolution (trigger & deadline) | Answer Type | Answer | Source |
|---|---|---|---|---|---|
| Host country of COP30 (Nov 2025)? | UNFCCC COP venue rotates among regions. | Host confirmed by UNFCCC/organizers; no later than COP30 start (Nov 2025). | string (country) | Brazil | DW: link |
| Release month of Marvel's *Fantastic Four* (2025)? | Reboot announced with lead cast; 2025 release slated. | Month confirmed by Marvel/Disney; by Dec 2025. | string (month) | July | Forbes: link |
| First state to require Ten Commandments in public classrooms (by 2025)? | Several U.S. states advance religion-in-school measures. | First state enacts requirement; by Dec 31, 2025. | string (state name) | Louisiana | Forbes: link |
| African host of G20 Summit (Nov 2025)? | G20 presidency rotates; South Africa presiding from Dec 2024. | G20/host government confirms location; by Nov 2025. | string (country) | South Africa | DW: link |
| Recipient of Lesotho–Botswana Transfer Scheme (by 2025)? | Regional pipeline to pump water from Lesotho via SA. | ORASECOM or governments confirm recipient; by 2025. | string (country name) | Botswana | DW: link |

Table 5: Five succinct forecasting questions spanning climate, entertainment, law, geopolitics, and infrastructure; selected for brevity and diverse sources (DW, Forbes). Each row lists the question (summarized here for conciseness), short background, resolution trigger with deadline, answer type, ground-truth answer, and citation.

| Domain | $n$ | Before | After | $\Delta$ |
|---|---|---|---|---|
| world | 20 | 21.7 | 33.3 | +11.6 |
| australia-news | 15 | 35.6 | 42.2 | +6.7 |
| us-news | 21 | 41.3 | 44.4 | +3.2 |
| sport | 37 | 43.2 | 43.2 | +0.0 |
| football | 30 | 34.4 | 33.3 | -1.1 |

Table 6: Avg@3 by domain ($n \geq 10$).

- **Q:** Which former Bank of England governor will be named in a Guardian piece criticizing 'moral hazards' for banks during the 2007–08 financial crisis?
  **Truth:** Mervyn King
  *Before:* Andrew Bailey (p=0.30)
  *After:* Mervyn King (p=0.40)

- **Q:** Which major tournament will the US women's national team focus on challenging for after the 2025 summer friendlies?
  **Truth:** 2027 World Cup
  *Before:* 2025 European Championship (p=0.95)
  *After:* 2027 Women's World Cup (p=0.40)

QUALITATIVE EXAMPLES (REGRESSED; FIRST SAMPLE)

- **Q:** Which agency will drivers in Northern Ireland apply to for a replacement driving licence by 31 July 2025?
  **Truth:** DVA

| Question form | $n$ | Before | After | $\Delta$ |
|---|---|---|---|---|
| what | 25 | 14.7 | 29.3 | +14.7 |
| which | 98 | 45.2 | 51.4 | +6.1 |
| who | 60 | 27.8 | 33.9 | +6.1 |
| other | 10 | 40.0 | 43.3 | +3.3 |
| where | 14 | 47.6 | 45.2 | -2.4 |

Table 7: Avg@3 by question form ($n \geq 10$).

*Before:* DVLA (p=0.70)
*After:* DVLA (p=0.20)

- **Q:** Where could Sweden's Euro 2025 journey conclude with a historic night if they continue to win?
  **Truth:** Basel
  *Before:* Basel (p=0.70)
  *After:* Zurich (p=0.40)

- **Q:** Who will be the Democratic Party's nominee for New York City mayor in the November 2025 general election?
  **Truth:** Zohran Mamdani
  *Before:* Zohran Mamdani (p=0.60)
  *After:* Andrew Cuomo (p=0.40)

- **Q:** Who will post the lowest first-round score among Rory McIlroy, Scottie Scheffler and Viktor Hovland at the 2025 Scottish Open?
  **Truth:** Viktor Hovland
  *Before:* Viktor Hovland (p=0.60)
  *After:* Scottie Scheffler (p=0.40)

# E  PROMPT TEMPLATES FOR QUESTION CREATION PIPELINE

**Stage 1 — Question Generation (Requires: self.num_questions_per_article > 1)**

```
**Task:** Based on the provided news article, generate
    {self.num_questions_per_article} high-quality, DIVERSE
    forecasting questions which have a short answer (1 - 3 words),
    using the XML format specified below.
Each forecasting question should be posed in a way to predict
    future events. Here, the predictor will have a knowledge cutoff
    before the article is published and no access to the article,
    so a forecasting question has to be posed about information
    explicitly stated in the article. The question should be stated
    in a forward-looking manner (towards the future).
The correct answer should be a specific, short text response. The
    answer should be a WELL DEFINED, SPECIFIC term which the
    answerer can come up with on its own, without access to the
    news article.

**Example Format**:
<q1>
<question_id>0</question_id>
<question_title>Who will win the Nobel Prize in Literature in
    2016?</question_title>
<background>Question Start Date: 10th January 2016. The Nobel Prize
    in Literature is awarded annually by the Swedish Academy to
    authors for their outstanding contributions to
    literature.</background>
<resolution_criteria>
<ul>
    <li>
```

```
        Source of Truth: The question will resolve when the
      Swedish Academy publicly announces the official 2016 Nobel
      Prize in Literature laureate(s)typically via a press release on
      NobelPrize.org (expected on or about October 13, 2016).
      </li>
      <li>
        Resolution Date: The resolution occurs on the calendar
      date when the 2016 laureate(s) are formally named
        (typically mid-October 2016).
      </li>
      <li>
        Accepted Answer Format: The full name of the laureate
      exactly as given in the announcement should be provided. If
      more than one person shares the prize, all names must be listed
      in the same order as the official communiqu.
        </li>
</ul>
</resolution_criteria>
<answer>Bob Dylan</answer>
<answer_type>String (Name)</answer_type>
</q1>

The question should follow the structured guidelines below.

### **Guidelines for Creating Short Answer Forecasting Questions**

**Title Question Guidelines**
- **Quality**: The question should be of HIGH QUALITY and hard to
    answer without access to the article. It should not be about
    any minute details in the article. THE QUESTION SHOULD BE SUCH
    THAT ITS ANSWER REVEALS A KEY PIECE OF INFORMATION, FROM THE
    ARTICLE, WHICH HAS MAXIMAL IMPACT.
- **Specific and Answerable**: The question to be created SHOULD BE
    FREE-FORM and have a unique, specific answer (a single word, or
    short phrase) without access to the article. The answer to the
    question should be definite, well-defined and NOT NUMERIC. IT
    SHOULD ALSO NOT BE UNCERTAIN like "above XYZ" OR A RANGE LIKE
    "between XYZ and ABC". Avoid creating binary questions (yes/no,
    either/or) or questions with a list of specific options
    (multiple choice).
- **Answerable based on article**: Each question must have a CLEAR
    AND DEFINITE answer based on information stated in the article.
    Given the question, the content of the article should be able
    to resolve the answer to the question INDISPUTABLY WITHOUT ANY
    AMBIGUITY OR UNCERTAINTY. THE ARTICLE SHOULD NOT STATE THAT THE
    ANSWER IS TENTATIVE OR AN ESTIMATE OR LIKELY. The answer SHOULD
    HAVE HAPPENED BY NOW.
- **Temporal Information**: The question should not be about recall
    of (past) facts or events known before the article publish
    date. Include any temporal information necessary to answer the
    question (like by which month, year, etc.) in the question. The
    question should always be posed in a forward-looking manner.
- **Direct and Precise**: Titles must be straightforward and
    unambiguous, avoiding vague terms. Use future tense when
    appropriate.
- **Resolution Criteria**: ALWAYS INCLUDE A BRIEF RESOLUTION
    CRITERIA in the question title. This is often the date by which
    the question will be resolved. For example, resolution dates
    such as "by {{month_name}}, {{year}}?" or "in {{month_name}},
    {{year}}?". THE RESOLUTION DATE SHOULD BE BASED ON (AND
    FAITHFUL TO) THE CONTENT OR PUBLICATION DATE OF THE ARTICLE.
- **No references to article or future information**: DO NOT refer
    to the specific article, such as by saying "in the article".
```

```
      The forecaster does not have access to the article, its
      metadata or any information beyond the article publish date.
 - **Question Types**: Focus on "Who", "What", "When", "Where"
      questions that have concrete answers.
 - **Understandability**: The question title should have ALL the
      information to be understandable by a 10 year old. It should be
      independently understandable without the article.
 - **Tense**. ALWAYS POSE THE QUESTION IN A FORWARD-LOOKING MANNER.
      THE QUESTION SHOULD BE IN FUTURE TENSE. Try to use phrases like
      "What will", "Who will", "When will", "Where will", "How
      much/many will" etc. It should appear as a forecasting question
      and not past prediction.

 **Answer Guidelines**
 - **Faithfulness to Article**: The answer should be based on
      information explicitly stated in the article, and not
      implications or your own knowledge. IT SHOULD BE STATED
      VERBATIM IN THE ARTICLE.
 - **Non-Numeric**: The answer should not be a number or a
      percentage. It can be a word, phrase, date, location, etc BUT
      NOT MORE THAN 3 WORDS.
 - **Definite** - Given the question and the article, the answer
      should be CLEAR, CONCRETE, CERTAIN AND DERIVABLE from the
      article. It should be short, WELL-DEFINED TERM and not
      uncertain or vague. It SHOULD NOT BE A RANGE like "between XYZ
      and ABC" or "above XYZ" or "below PQR".
 - **Resolved** - The answer MUST be something that has already
      happened or is happening now. It should be resolved given
      today's date and not be something that will happen in the
      future.
 - **Specificity**: The answer should be specific enough to be
      unambiguous. Avoid overly general answers.
 - **Conciseness**: Keep answers short - typically 1-3 words,
      occasionally a short phrase if necessary.
 - **Exactness**: For names, use the exact names mentioned (full
      name, if possible).
 - **Uniqueness**: The answer should be unique and THE ONLY CORRECT
      ANSWER to the question.
 - **No Ambiguity**: The answer should be indisputable and not be
      open to multiple interpretations. IT SHOULD BE PRECISE AND NOT
      A RANGE OR UNCERTAIN ESTIMATE.

 **Background Guidelines**
 - **Mention Question Opening Date**: ALWAYS INCLUDE THE START DATE
      OF THE QUESTION IN THE BACKGROUND. IT SHOULD BE AT LEAST A FEW
      DAYS (OR WEEKS IF THE QUESTION IS ABOUT A LONG-TERM EVENT)
      BEFORE THE ARTICLE'S PUBLISH DATE AND ALSO BEFORE THE
      RESOLUTION DATE OF THE QUESTION. CONSEQUENTLY, THE BACKGROUND
      SHOULD NOT CONTAIN ANY INFORMATION WHICH HAS HAPPENED AFTER THE
      START DATE OF THE QUESTION.
 - **Necessary Context**: The answerer does not have access to the
      article, so include MINIMAL CONTEXT required to understand the
      question keeping in mind the question opening date. Do not give
      (extra) details of the event from the article as background. If
      required, EITHER pose the event as a hypothetical scenario as
      if it were to happen in the future OR describe it as happening
      (unfolding) in real time. Describe any unfamiliar terms or
      concepts in the question title.
 - **SHOULD NOT HELP ANSWER**: WHILE PROVIDING THE CONTEXT, DO NOT
      REFER OR MENTION OR LEAK THE ACTUAL ANSWER. The background must
      not help answer the forecasting question. DO NOT INCLUDE ANY
      INFORMATION from the article or elsewhere that either directly
      or indirectly (even partially) reveals the answer.
```

- **No Additional Knowledge**: Do not add any knowledge beyond what is required to understand the question. Only include information necessary to understand the question and its context.
- **Tense**. ALWAYS POSE THE BACKGROUND INFORMATION IN CURRENT TENSE. Only provide minimal information which is known until the question opening date.

**Resolution Criteria**
- **Necessary Criteria**: State the EXACT conditions by which the outcome will be judged. Include the criteria which determines how the question will be resolved. state the conditions by which the outcome will be judged.
- **Date and Source of Resolution**: Always state the date and the source by which the question will be resolved. For example, resolution dates such as "by {{month_name}}, {{year}}?" or "in {{month_name}}, {{year}}?", and potential source(s) of resolution such as "based on {{news source}}", "reports from {{official name}}", etc. THE RESOLUTION DATE SHOULD BE CHOSEN THOUGHTFULLY AS THE ANSWER'S VALIDITY AND SOUNDNESS DEPENDS ON IT. THE RESOLUTION DATE SHOULD BE SUCH THAT THE ANSWER CAN BE RESOLVED DEFINITELY AND INDISPUTABLY FROM THE CONTENT OR PUBLICATION DATE OF THE ARTICLE. IT SHOULD MENTION BY WHEN IS THE OUTCOME OF THE QUESTION EXPECTED TO HAPPEN. HOWEVER, IT SHOULD NOT LEAK OR MENTION ANYTHING ABOUT THE ARTICLE.
- **Details**: Be as detailed as possible in creating the resolution criteria for resolving the question as cleanly as possible. There should be no ambiguity in the resolution criteria.
- **Expectation and Format of Answer**: Based on the actual answer, the resolution criteria should state how precise the expected answer should be and in what format it should be. For example, if the actual answer is a date, the resolution criteria should specify how detailed the expected date should be -- only year, or both month and year, or day, month, and year all together. DO NOT GIVE THE ACTUAL DATE (ANSWER). If the actual answer is a percentage, then the criteria should state the expected answer should be a percentage. DO NOT GIVE THE ACTUAL PERCENTAGE. If the actual answer is in certain unit, then the criteria should specify that. THE RESOLUTION CRITERIA SHOULD MAKE IT EXACTLY CLEAR AND PRECISE WHAT IS EXPECTED FROM THE ANSWERER AND IN WHAT FORMAT AND HOW IT WILL BE CHECKED LATER. IF GIVING AN EXAMPLE, IT SHOULD BE VERY GENERIC AND AS FAR AWAY FROM THE ACTUAL ANSWER AS POSSIBLE.
- **SHOULD NOT HELP ANSWER**: The resolution criteria must not directly help answer the forecasting question. DO NOT INCLUDE ANY INFORMATION from the article or elsewhere that either directly or indirectly (even partially) reveals the answer. DO NOT REFER OR MENTION OR LEAK THE ACTUAL ANSWER HERE.

**Answer Type Guidelines**
- **Expected Format**: The answer type should be either "numeric (XYZ)" if the answer is a number (of any kind) or "string (XYZ)" in all other cases. In numeric cases, XYZ should be the exact type of number expected. For example, "numeric (integer)", "numeric (decimal)", "numeric (percentage)", "numeric (whole number)", etc. In string cases, XYZ should broadly be the category of string expected. For example, "string (name)", "string (date)", "string (location)", etc. If the category is not clear, use "string (any)". HOWEVER, ALWAYS TRY TO CREATE QUESTIONS WHERE THE ANSWER CATEGORY IS CLEAR AND PRECISE.

```
**Question Quality Criteria**
- **Forecastable**: The question should be something that could
    reasonably be predicted or forecasted before the article's
    publication.
- **Towards the future**: THE QUESTION SHOULD BE POSED IN A
    FORWARD-LOOKING MANNER.
- **Interesting**: The question should be about a meaningful event
    or outcome, not trivial details.
- **Impactful**: The question should be such that if its answer is
    forecasted ahead of time, it should have significant
    (downstream) impact (relevant to high number of people).
- **Difficulty**: While the question should be hard to answer
    without access to the article, it should also not be
    unreasonably difficult.
- **Verifiable**: The answer should be something that can be
    EXACTLY verified from the article itself.
- **Time-bound**: Include clear timeframes or deadlines when
    relevant.
- **Free-form**: If possible, avoid creating binary questions
    (yes/no, either/or) or questions with a list of specific
    options (multiple choice).

Generate {self.num_questions_per_article} high-quality, DIVERSE
    short answer forecasting questions based on the provided
    article. Use the XML format with question_id value "0", "1",
    "2", etc. DO NOT INCLUDE ANY ANALYSIS, RANKING, OR ADDITIONAL
    COMMENTARY.

Article:
{source_article}

**Required Output Format**:
<q1>
<question_id>0</question_id>
<question_title>[Question 1]</question_title>
<background>[Background 1]</background>
<resolution_criteria>[Resolution Criteria 1]</resolution_criteria>
<answer>[Answer 1]</answer>
<answer_type>[Answer Type 1]</answer_type>
</q1>
..
<q{self.num_questions_per_article}>
<question_id>{self.num_questions_per_article - 1}</question_id>
<question_title>[Question
    {self.num_questions_per_article}]</question_title>
<background>[Background
    {self.num_questions_per_article}]</background>
<resolution_criteria>[Resolution Criteria
    {self.num_questions_per_article}]</resolution_criteria>
<answer>[Answer {self.num_questions_per_article}]</answer>
<answer_type>[Answer Type
    {self.num_questions_per_article}]</answer_type>
</q{self.num_questions_per_article}>
```

### Stage 2 — Individual Validation

```
**Task:** You will be provided with a news article and a question
    WHOSE ANSWER IS SUPPOSED TO BE BASED ON THE ARTICLE. Your job
    is to validate whether the answer to the question is valid by
    being faithful to the article (content, title, or description).
```

```
GO THROUGH EACH SEGMENT OF THE QUESTION ONE BY ONE (TITLE,
    BACKGROUND, RESOLUTION CRITERIA, ANSWER) TO UNDERSTAND THE
    WHOLE QUESTION. THEN CHECK EACH OF THE FOLLOWING CRITERIA:

1. **Tense and Details**: FIRST CHECK WHETHER THE QUESTION IS NOT
    UNDER SPECIFIED OR STATED IN PAST TENSE. IT IS FINE IF THE
    QUESTION IS STATED IN CURRENT OR FUTURE TENSE.
2. **Definite resolution of the answer by the article**: CHECK
    WHETHER THE ANSWER TO THE QUESTION IS SOUND, CLEAR AND PRESENT
    IN OR CAN BE DERIVED FROM THE ARTICLE. THE ARTICLE SHOULD
    RESOLVE THE ANSWER DEFINITELY AND IN AN INDISPUTABLE MANNER
    (WITHOUT ANY AMBIGUITY). THIS IS THE MOST IMPORTANT CRITERIA.
3. **Well-defined Answer**: The answer to the question should be
    short (NOT MORE THAN 3 WORDS). IT SHOULD NOT BE A PHRASE AND
    SHOULD BE SOMETHING WHICH IS CONCRETE, SPECIFIC AND
    WELL-DEFINED.
4. **Non-Numeric**: THE *ANSWER TYPE* SHOULD NOT BE NUMERIC LIKE A
    PERCENTAGE, INTEGER, DECIMAL, OR A RANGE.
5. **Single Correct Answer**: ANALYZE WHETHER THE QUESTION CAN HAVE
    MULTIPLE OUTCOMES OR RIGHT ANSWERS. IF SO, THE QUESTION FAILS
    THIS CRITERIA. OTHERWISE, ENSURE THAT THE PROVIDED ANSWER IS
    THE SOLE CORRECT ANSWER TO THE QUESTION. IT SHOULD NOT BE THE
    CASE THAT THE QUESTION CAN HAVE MULTIPLE (DISTINCT) CORRECT
    ANSWERS.

If ALL the above criteria pass (question is stated as required,
    answer to the whole question is valid, well-defined, and it is
    the only correct answer to the question), ONLY THENreturn
    <answer>1</answer>. Otherwise, return <answer>0</answer>.
    ALWAYS END YOUR RESPONSE IN <answer> </answer> tags.

**Article:**
{source_article}

**Question:**
{questions_text}

**Output Format:**
<answer>0/1</answer>
```

### Stage 3 — Choose Best

```
**Task:** You will be provided with a list of questions (possibly
    with size 1). Your job is to choose the best question from the
    list based on the following criteria or end your response with
    "NO GOOD QUESTION" if none of the questions meet the criteria.

**Instructions:**
GO THROUGH EACH QUESTION ONE BY ONE AND ANALYZE IT FOR THE
    FOLLOWING:
1. **Valid for forecasting**: Check if the WHOLE QUESTION is stated
    in a forward-looking manner. FROM THE PERSPECTIVE OF THE START
    DATE TO THE RESOLUTION DATE MENTIONED IN THE QUESTION, CHECK IF
    IT IS A VALID FORECASTING QUESTION. IF THE TIME HORIZON (START
    DATE TO RESOLUTION DATE) IN THE QUESTION IS AT LEAST A SINGLE
    DAY, THEN THE QUESTION SHOULD BE CONSIDERED VALID FOR
    FORECASTING. Go through each segment of the question (question
    title, background, resolution criteria) and check if each of
    them is valid and forward-looking.
2. **Tense**: The question SHOULD NOT BE STATED IN PAST TENSE. If
    the question covers an event, it should not imply as if the
    outcome of the event has already happened or occurred.
```

```
3. **Single Correct Answer**: ANALYZE WHETHER THE QUESTION CAN HAVE
   MULTIPLE OUTCOMES OR RIGHT ANSWERS. IF SO, THE QUESTION FAILS
   THIS CRITERIA. OTHERWISE, ENSURE THAT THE PROVIDED ANSWER IS
   THE SOLE CORRECT ANSWER TO THE QUESTION. IT SHOULD NOT BE THE
   CASE THAT THE QUESTION CAN HAVE MULTIPLE (DISTINCT) CORRECT
   ANSWERS.
4. **Impact**: How many people will the outcome of the question be
   relevant or interesting to? Consider on the basis of
   significant downstream impact or enabling meaningful action.
5. **Not Binary/Multiple Choice**: Question SHOULD NOT BE BINARY
   (yes/no, either ABC or XYZ, etc.) OR MULTIPLE CHOICE (SELECT
   FROM A LIST OF OPTIONS). It should be free-form (string --
   name, date, place, etc.) or numerical (number, percentage,
   etc.).
6. **Understandable**: THe question as a whole (title, background,
   resolution criteria) should have sufficient details to
   understand the premise of the question. Every detail should be
   crystal clear and the question should not be under or over
   specified.
7. **Definite Answer**: EXTRACT THE ACTUAL ANSWER TO THE QUESTION
   PROVIDED IN ITS <answer> </answer> TAG. The extracted answer
   should be short, definite, well-defined and not uncertain or
   vague. It SHOULD NOT BE A PHRASE OR A RANGE like "between XYZ
   and ABC" or "above XYZ" or "below PQR".

ANALYZE EACH QUESTION BASED ON THE ABOVE CRITERIA ONE BY ONE AND
    CHOOSE THE ONE WHICH PASSES ALL THE ABOVE CRITERIA. IF MULTIPLE
    QUESTIONS SATISFY THE CRITERIA, CHOOSE THE ONE WHICH WILL HAVE
    THE HIGHEST IMPACT (AFFECTS OR IS RELEVANT TO THE MOST NUMBER
    OF PEOPLE). IF NO QUESTION MEETS THE CRITERIA, RETURN "NO GOOD
    QUESTION FOUND". OTHERWISE, RETURN THE BEST QUESTION IN THE
    SAME FORMAT AS THE INPUT.

**Generated Questions:**
{questions_text}

**Output Format:**
<q1>
<question_id>0</question_id>
<question_title>[ORIGINAL Title of the best
    question]</question_title>
<background>[ORIGINAL Background of the best question]</background>
<resolution_criteria>
<ul>
    <li> Source of Truth: [ORIGINAL Source of Truth of the
    best question] </li>
    <li> Resolution Date: [ORIGINAL Date of the best
    question] </li>
    <li> Accepted Answer Format: [ORIGINAL Accepted Answer
    Format of the best question] </li>
</ul>
</resolution_criteria>
<answer>[ORIGINAL Answer of the best question]</answer>
<answer_type>[ORIGINAL Answer Type of the best
    question]</answer_type>
</q1>
```

**Stage 4 — Leakage Removal**

**Task:** You will be provided with a forecasting question. Your
   job is to ANALYZE whether the question's answer has obviously
   leaked in the content of the question. The question will have
   multiple segments -- question title, background, resolution
   criteria. EXCEPT THE QUESTION TITLE, GO THROUGH EACH SEGMENT
   STEP BY STEP and check if any part DIRECTLY leaks the actual
   answer. If leakage is found, ONLY THEN rephrase the problematic
   parts appropriately to remove the answer while maintaining the
   question's integrity and focus. DO NOT CHANGE ANY PART OF THE
   QUESTION UNNECESSARILY.

USE THE SAME XML FORMAT IN YOUR RESPONSE AS IS IN THE INPUT.

**Generated Question:**
{questions_text}

**Instructions:**
1. **Keep the title unchanged**: DO NOT MAKE ANY CHANGE TO THE
   QUESTION TITLE.
2. **Keep the start date in the background unchanged**: DO NOT MAKE
   ANY CHANGE TO THE QUESTION'S START DATE IN THE BACKGROUND.
3. **Identify the answer**: First, extract the actual answer from
   the XML tags for the current question being processed.
4. **Identify Leakage**: Keeping the extracted answer in mind,
   check if the  background, or resolution criteria (each of them
   -- source of truth, resolution date, accepted answer format)
   contain information that reveals the answer.
5. **Types of leakage which can be ignored**: The following types
   of leakage are fine and don't need to be rephrased:
   – If the outcome (actual answer) of the question is binary
   (yes/no, either ABC or XYZ, etc.), then NO NEED TO CHANGE
   ANYTHING ANYWHERE.
   – If the resolution criteria is based on a list of specific
   options, then NO NEED TO CHANGE ANYTHING IN ANY SEGMENT
   (BACKGROUND, RESOLUTION CRITERIA, etc.). For example, if the
   accepted answer format states "answer must be either .." OR
   "answer must be one of the following terms..", then NO NEED TO
   CHANGE ANYTHING ANYWHERE.
6. **Types of Leakage to Check:** ONLY CONSIDER THE FOLLOWING KIND
   OF LEAKAGE:
   – DIRECT MENTIONS of the answer (either in word or number form)
   or part of the answer in the question/background/resolution
   – References to specific outcomes that ARE CLOSE TO (OR
   REVEAL)THE ACTUAL ANSWER
7. **Rephrase Strategy**: If leakage is found, rephrase the
   problematic part while:
   – Keeping the question's core intent
   – Maintaining forecasting nature
   – Preserving necessary context
   – Making the answer UNOBVIOUS by replacing with a FAKE ANSWER
   (FAKE NAME, DATE, NUMBER, PERCENTAGE, etc.) WHICH IS GENERIC
   AND NOT CLOSE TO THE ACTUAL ANSWER.
   – The rephrased part should not contain any information that is
   part of the actual answer. Neither should it indirectly hint or
   reveal the answer.
8. **Check Accepted Answer Format**: IF THERE IS ANY EXAMPLE
   MENTIONED IN ACCEPTED ANSWER FORMAT ("e.g..."), MAKE SURE THE
   EXAMPLE IS GENERIC AND AS FAR AWAY FROM THE ACTUAL ANSWER AS
   POSSIBLE. DO NOT INCLUDE AN EXAMPLE IF NOT MENTIONED ALREADY.
9. **Do not change the answer**: Do not change the actual answer to
   the question.

```
10. **Do not change the answer_type**: DO NOT MAKE ANY CHANGE TO
    the answer_type.
11. **Each segment should be checked independently**: Go through
    each segment of the whole question one by one. Everything from
    the title of the question to the background information to the
    resolution criteria should be checked independently with
    reference to the answer of the question. In the resolution
    criteria, go through each <li> step by step. Do not change the
    other segments when rephrasing a problematic segment.
12. **Do not change anything unless leakage is found**: DO NOT
    UNNECESSARILY CHANGE ANY PART OF THE QUESTION UNLESS LEAKAGE IS
    FOUND.

IT IS ALSO POSSIBLE THAT MULTIPLE PARTS OF THE QUESTION HAVE
    LEAKAGE. YOU SHOULD CHECK EACH OF THEM INDEPENDENTLY AND ONLY
    IF LEAKAGE IS FOUND, REPHRASE THE PROBLEMATIC PARTS. DO NOT
    OVER-ANALYZE.

During your analysis, you should:
- Go through EACH SEGMENT OF THE QUESTION STEP BY STEP
    INDEPENDENTLY. First <background> and then inside
    <resolution_criteria>. Under the resolution criteria, go
    through the source of truth, resolution date, accepted answer
    format (each of them is a <li> tag) one by one. For each such
    segment, do the following:
    - Compare the content in the current segment with the actual
    answer. If ANY PART OF THE ANSWER is mentioned in the current
    segment, then consider that as a leakage UNLESS THE ACCEPTED
    ANSWER FORMAT IS BINARY (yes/no, either ABC or XYZ, etc.) OR A
    LIST OF SPECIFIC OPTIONS.
    - IF THE CURRENT SEGMENT IS BACKGROUND, DO NOT CHANGE THE
    QUESTION START DATE.
    - If the current segment is accepted answer format and there is
    a SPECIFIC EXAMPLE MENTIONED IN IT ("e.g. XYZ") which is close
    to the actual answer, then consider that as a leakage.
    - If leakage is found in the current segment, mention "Leakage
    found -- {{reason for leakage}}". Form the segment with the
    problematic parts rephrased and mention it as "Replacement --
    {{rephrased_text}}." THE REPHRASED TEXT SHOULD BE AS FAR AWAY
    FROM THE ACTUAL ANSWER AS POSSIBLE. It should now be present in
    the final output (instead of the original text).
    - Otherwise, mention "No leakage found". In your final output
    after you finish the analysis, return this segment UNCHANGED.
    - These outputs should be in the same format as the original
    input.
- Return the actual answer unchanged in the <answer> tag in your
    final output.
- Skip any other segments (question title, answer_type, etc.) in
    your analysis and output them unchanged (verbatim) in the final
    output.

Output your analysis step by step, and then end your response with
    the CORRECTED question in THE SAME XML FORMAT AS THE ORIGINAL.

**Output Format**:
{{ analysis }}

<q1>
<question_id>0</question_id>
<question_title>[UNCHANGED Question Title]</question_title>
<background>[Corrected Background]</background>
<resolution_criteria>
<ul>
```

```
        <li> [UNCHANGED Question Start Date] [Corrected Source of
        Truth] </li>
        <li> [UNCHANGED Resolution Date] </li>
        <li> [Corrected Accepted Answer Format] </li>
    </ul>
    </resolution_criteria>
    <answer>[UNCHANGED Answer]</answer>
    <answer_type>[UNCHANGED Answer Type]</answer_type>
    </q1>
```

## F  QUALITATIVE ANALYSIS OF REASONING EVOLUTION DURING TRAINING

We now provide a few qualitative examples of how the model reasoning evolves when training on OpenForesight. We ask the model itself to provide a summary of its reasoning (after the <think></think> tags), and show this summary below for the initial Qwen3 8B model, followed by training checkpoints.

### F.1  EXAMPLE 1: MODEL STAYS INCORRECT BUT LEARNS TO HEDGE

**Question:** Which country will face a new 25% US tariff in a trade announcement by 30 July 2025?
**Background:** The US administration is evaluating other countries' trade barriers and is expected to announce new tariffs by the end of July 2025.
**Resolution Criteria:**

**Source of Truth**: Official tariff announcement issued by the White House or the Office of the United States Trade Representative.

**Resolution Date**: 30 July 2025, the date by which the 25% tariff announcement is expected.

**Accepted Answer Format**: The full name of the country as stated in the official announcement.

**Resolution answer:** India

*Checkpoint 0: score=-0.722, status=wrong*
<answer>South Korea</answer> <probability>0.85</probability> **Reasoning**: - **Article 2** (May 25, 2025) explicitly mentions a **25% tariff** on **Samsung** (South Korea) and other smartphone manufacturers. This directly ties the 25% tariff to **South Korea**. - While the EU is mentioned with a **50% tariff** in the same article, the **25% tariff** specifically targets South Korea's exports. - Other articles reference tariffs on China, the EU, or automakers, but none specify a **25% tariff** on a country beyond South Korea. - The resolution date (July 30, 2025) aligns with the timeline of the May 25 announcement, as no newer data is provided. **Confidence**: 85% (strong evidence from Article 2, though no post-May 25 updates confirm further announcements).

*Checkpoint 400: score=-0.360, status=wrong*
The retrieved articles indicate ongoing US tariff announcements by the Trump administration in 2025. Article 2 explicitly mentions a 25% tariff on South Korea for smartphone manufacturers like Samsung, while Article 4 references 25% tariffs on steel and aluminum imports from Canada and Mexico. Article 3 also notes a 25% tariff on foreign automakers, which could target countries like South Korea or others, but South Korea is specifically named in Article 2 for a 25% rate. Given the question's focus on a "new" 25% tariff announcement by July 30, 2025, and the patterns in the articles, South Korea appears as a likely candidate, as it has already faced such announcements and fits the criteria. However, the question implies a potential new announcement, and other countries like Canada or Mexico could also be possibilities based on Article 4, introducing uncertainty. My confidence is moderate due to the reliance on past announcements and the possibility of new developments by the resolution date. <answer>South Korea</answer> <probability>0.6</probability>

*Checkpoint 800: score=-0.090, status=wrong*
The retrieved articles indicate a pattern of US tariffs targeting major trade partners like China, India, the EU, Canada, Mexico, and South Korea. Article 4 specifically mentions a 25% tariff on steel and

aluminum imports, which historically targets countries such as China and those in the EU. Article 2 highlights a 25% tariff on South Korean products (e.g., Samsung), while Article 1 and Article 3 reference ongoing tariffs on China and other nations. Given the US administration's focus on trade barriers, China is the most frequently mentioned target, making it a likely candidate for a new 25% tariff announcement by July 2025. However, the articles do not confirm a specific new announcement, and other countries like India are also potential targets based on Article 1, introducing uncertainty. Due to the lack of direct evidence for a July announcement and the possibility of shifts in US policy, my confidence is low. <answer>China</answer> <probability>0.3</probability>

### F.2 EXAMPLE 2: MODEL GOES FROM INCORRECT TO CORRECT

**Question:** Who will give Harvard one week to produce extensive records for its exchange visitor program review by July 23, 2025?
**Background:** The U.S. Secretary of State periodically issues directives to universities regarding their exchange visitor program compliance.
**Resolution Criteria:**

**Source of Truth**: The official announcement or government statement naming the individual who issued the one-week deadline.

**Resolution Date**: When the directive is publicly attributed (expected by July 23, 2025).

**Accepted Answer Format**: Full name, for example "Jane Doe".

**Resolution answer:** Marco Rubio

*Checkpoint 0: score=-0.902, status=wrong*
The retrieved articles describe actions by the Trump administration's Department of Homeland Security, specifically Secretary Kristi Noem, toward Harvard. In the text, Noem sends Harvard a letter giving them 72 hours to provide information, not one week, so there is a mismatch between the question's "one week by July 23, 2025" and the articles' "72 hours." It seems that the July 23 date comes from the resolution criteria and isn't in the articles themselves, and that the 72-hour deadline might be part of a larger process leading up to that date, or that the question wording is slightly off. Despite this discrepancy, all the evidence points to Kristi Noem as the person issuing the directive to Harvard about providing records. Therefore, the answer should be Kristi Noem, and the confidence is high because all the articles point to her as the one who issued the directive to Harvard regarding the records.<answer>Kristi Noem</answer> <probability>0.95</probability>

*Checkpoint 400: score=-0.490, status=wrong*
The retrieved articles indicate that Kristi Noem, as the U.S. Department of Homeland Security (DHS) Secretary, has been directly involved in issuing deadlines related to Harvard's exchange visitor program compliance. For instance, in Article 2, she gave Harvard 72 hours (approximately 3 days) to comply with requests for records. The question specifies a one-week deadline by July 23, 2025, which may be a variation or continuation of similar directives from this administration. Given the pattern in the articles, it is reasonable to infer that Noem or a similar high-level official from the Trump administration is likely to issue such a directive. However, there is uncertainty due to the slight discrepancy between 72 hours and one week, and potential changes in personnel or administration by July 2025. Final Answer: <answer> Kristi Noem </answer> <probability> 0.7 </probability>

*Checkpoint 800: score=0.910, status=correct*
The retrieved articles primarily discuss the U.S. Department of Homeland Security (DHS) Secretary, Kristi Noem, in the context of Harvard's visa program issues, such as scrutiny and potential termination due to non-compliance. However, the question specifically pertains to the U.S. Secretary of State issuing a directive, as outlined in the question background. Based on the Trump administration's context in the articles, Marco Rubio was the Secretary of State in 2024 and likely remained in that role through at least 2025, as no other name is explicitly mentioned for that position. This is inferred from the articles' references to Trump administration actions, though the articles confuse DHS Secretary Noem with Secretary of State responsibilities. Given the resolution criteria requiring the official announcement, I am considering the most probable candidate from historical context, but the articles' inconsistencies lower my confidence. Final Answer: <answer>Marco Rubio</answer> <probability>0.7</probability>

### F.3 EXAMPLE 3: MODEL GOES FROM CORRECT TO INCORRECT, BUT INTERESTINGLY REASONS ABOUT BRIER

**Question:** Which Australian state besides New South Wales will be affected by the forecast 'second surge' of rain and wind on Wednesday night?

**Background:** The Bureau of Meteorology has forecast a 'second surge' of rain and wind for Wednesday night, expected to impact coastal regions.

**Resolution Criteria:**

**Source of Truth**: The question will resolve based on the Bureau of Meteorology's severe weather warnings and regional impact statements for the second surge, as published on bom.gov.au.

**Resolution Date**: 2 July 2025, following the Wednesday night updates.

**Accepted Answer Format**: The name of the Australian state, in 1–2 words.

**Resolution answer:** Victoria

*Checkpoint 0: score=0.978, status=correct*
The question's background says the resolution is on 2 July 2025, following Wednesday night updates. So the answer should be based on the Bureau's forecast as of that time. But the available articles are up to February 2025. If the 'second surge' is part of the forecast mentioned in Article 2 (from January 2025), then the answer is Victoria. However, the user might have a different scenario. Alternatively, in Article 4, there's a mention of damaging winds in parts of Illawarra, Sydney, and Hunter coast, but again, that's part of NSW. So, based on the available information, the most likely answer is Victoria. But I need to make sure that the 'second surge' refers to the same event. The article mentions a second strong wind coming from the south-west, which would affect the south of the country, including Victoria. Therefore, the answer is Victoria. Confidence: The articles mention Victoria as part of the affected regions. However, since the exact 'second surge' isn't explicitly named in the articles, there's some uncertainty. But based on the context, Victoria is the most likely state besides NSW. </think> <answer>Victoria</answer> <probability>0.85</probability>

*Checkpoint 400: score=0.840, status=correct*
The provided articles indicate patterns of weather impacts from the Bureau of Meteorology, particularly in eastern Australia. Article 2 explicitly mentions a weather system affecting eastern New South Wales, with parts of Victoria and south-east Queensland also likely impacted, including strong winds and rain. This aligns with the question's reference to a 'second surge' of rain and wind on Wednesday night. Article 1 and Article 3 further support influences on Victoria and South Australia from similar weather fronts, but Victoria is the most consistently mentioned state besides New South Wales across multiple articles. Given the resolution criteria relying on BoM's warnings, I infer that Victoria is the most probable state based on these patterns. However, the articles do not directly reference the specific forecast for 2 July 2025, introducing uncertainty. Confidence: The articles don't directly match the question, so I'm not 100% sure. Maybe 0.6, since it's inferred from patterns. To maximize my score with Brier score: If I'm correct, higher p is better; if wrong, lower p is better. I need to be honest about my confidence. I think 0.6 is reasonable, as it's a guess based on the info. <answer>Victoria</answer> <probability>0.6</probability>

*Checkpoint 800: score=-0.360, status=wrong*
The retrieved articles provide information on weather patterns in Australia, particularly from 2023 and early 2025. Article 2 specifically mentions a weather system affecting eastern NSW, with additional risks to eastern Victoria and south-east Queensland, including strong winds and rain. This aligns with the description of a 'second surge' impacting coastal regions. Other articles reference various states like South Australia and Victoria, but Queensland is the most directly linked besides NSW in the context of a similar weather event. Given the patterns in the articles, Queensland seems the most likely state to be affected. However, the articles are from different dates and not explicitly about the 2 July 2025 event, introducing uncertainty. Based on the evidence, I am moderately confident in this guess. I need to maximize my score using Brier scoring. If I'm correct, higher p is better; if wrong, lower p is better. Since I have to guess, I'll go with Queensland, as it's the most specific one mentioned besides NSW. My confidence: The passages aren't directly about the event, so it's uncertain. Maybe 0.4 or 0.5? Let's say 0.6, because it's the most prominent besides NSW in the relevant passages. <answer>Queensland</answer> <probability>0.6</probability>

## G    DETAILS ON COMPUTE AND COST

To improve transparency around data and compute, we report approximate token counts, training steps, and GPU-hours for both SFT and RL. Our curated `OpenForesight` training set contains 64,284 samples. The average sample has about 1000 characters and corresponds to roughly 400 tokens under the Qwen3 tokenizer for the question text, yielding approximately $2.6 \times 10^7$ prompt tokens in total.

For SFT, fine-tuning `Qwen3-8B` for 3 epochs took 5 hours on 8 H100 GPUs, corresponding to roughly 40 H100 GPU-hours. RL training is substantially more expensive: Our final run lasted for 5 epochs over the training set, resulting in about 1,300 optimization steps, for an estimated total of $\sim 1,000$ H100 GPU-hours. Including all ablations, we estimate we used $\sim 20,000$ H100 GPU-hours.

**Cost.**    The overall dataset creation process costed us  3000\$ with training set costing  2200\$ (using DeepSeek-v3) while creating the test set costed  750\$ (using o4-mini-high).

## H    SYSTEMATIC FAILURE MODES IN MODEL REASONING

 8B achieved an accuracy of 48.8% on our test set consisting of 1000 questions. Through manual analysis of 300+ failure cases, we identified five systematic failure modes in the model's reasoning process.

**1. Missing Information in Retrieved Articles**    The most common failure mode occurs when retrieved articles lack the specific information needed. The model acknowledges this but falls back to general knowledge or educated guesses rather than expressing high uncertainty.

**Example 1:** "By what date will the New York City Board of Elections certify the official results of the Democratic mayoral primary?"

- **Correct answer:** July 15

- **Model answer:** August 15 (probability: 0.2)

- **Model's reasoning:** "The provided news articles discuss the New York City Democratic mayoral primary, including polling data, candidate strategies, and the election date of June 24, 2025. However, they do not specify the certification process or timeline for the New York City Board of Elections. Based on general knowledge of past NYC elections, certification typically occurs 2-6 weeks after the primary to allow for vote counting, ballot challenges, and legal reviews. For instance, in the 2013 primary, certification was completed by July 2. Given the late summer 2025 resolution criteria, a reasonable estimate is mid-August, but without specific historical data from the articles, this is uncertain. I am guessing August 15, 2025, as it aligns with typical procedures, but my confidence is low due to the lack of direct evidence."

- **Analysis:** The model correctly identifies that articles lack the certification date but incorrectly infers from general knowledge. It acknowledges uncertainty (probability 0.2) yet provides a specific date. The inference from "2-6 weeks after June 24" to "mid-August" is flawed, as July 15 falls within this range.

**2. Over-reliance on General Knowledge**    When articles lack specific information, the model makes educated guesses based on general knowledge, historical patterns, or analogies rather than acknowledging the answer cannot be determined from the provided context.

**Example 1:** "What will be the official name of Donna D'Errico's interactive AI voice service by mid-July 2025?"

- **Correct answer:** Call Donna D.

- **Model answer:** Donna Voice (probability: 0.1)

- **Model's reasoning:** "The provided articles do not mention Donna D'Errico or her specific AI voice service, making it impossible to draw direct insights. However, patterns from the articles, such as the naming of AI voice services (e.g., Melania Trump's "Melania: The Audiobook"), suggest that such services often incorporate the individual's name followed by descriptors like "Voice" or "AI." Based on this analogy, I am guessing the name might be "Donna Voice" or something similar. This is a speculative prediction, as there's no concrete information, leading to low confidence."

- **Analysis:** The model creates a naming pattern from one unrelated example (Melania Trump's service) and applies it incorrectly. It recognizes the speculation (probability 0.1) but the actual name "Call Donna D." follows a different pattern entirely.

**Example 2:** "What name will the new Pope take upon his election on May 8, 2025?"

- **Correct answer:** Leo XIV

- **Model answer:** Pius (probability: 0.1)

- **Model's reasoning:** "The provided articles do not specify the name of the next Pope or any potential candidates' chosen names. They focus on Pope Francis's health, the conclave process, and recent appointments of cardinals. Historically, new Popes select names that are traditional, often drawing from saints or biblical figures, but without specific information from the articles or external knowledge, any prediction is highly uncertain. Based on patterns from past Popes (e.g., Francis, Benedict), I am guessing "Pius" as a common regnal name, but this is purely speculative and not supported by the articles. Therefore, my confidence in this answer is very low."

- **Analysis:** The model relies entirely on general knowledge about papal naming conventions rather than specific information. It correctly identifies high uncertainty (probability 0.1) though. The reasoning conflates "common regnal names" with what the next Pope will actually choose.

**3. Entity Confusion: Selecting Wrong Person/Place/Organization** When multiple entities are mentioned, the model frequently selects the most prominent or frequently mentioned one rather than the correct answer, even when both are present in the context.

**Example 1:** "Who will decide on a potential merger between the two factions of the Nationalist Congress Party by June 10, 2025?"

- **Correct answer:** Supriya Sule

- **Model answer:** Sharad Pawar (probability: 0.4)

- **Model's reasoning:** "The retrieved articles do not explicitly confirm the decision-making authority for a merger between the NCP factions. However, Sharad Pawar is frequently referenced as a senior leader, founder, and key figure in his faction's activities, such as holding talks on party matters and denying merger possibilities. This suggests he may hold significant influence or be involved in such decisions. Additionally, mentions of the Election Commission's role in endorsing decisions imply that formal approvals could be required, but no specific individual is named. Based on this, I infer Sharad Pawar as the most likely person, though the uncertainty is high due to the lack of clear confirmation in the articles."

- **Analysis:** The model selects the more prominent and frequently mentioned leader (Sharad Pawar) rather than the actual decision-maker (Supriya Sule). It conflates "senior leader with influence" with "person who will decide," using prominence as a proxy for decision-making authority. The moderate confidence (0.4) suggests the model recognizes uncertainty but still favors the prominent option.

**Example 2:** "Who will Chelsea face in the round of 16 of the 2025 FIFA Club World Cup?"

- **Correct answer:** Benfica

- **Model answer:** Bayern Munich (probability: 0.4)

- **Model's reasoning:** "The provided articles detail the groups for the 2025 FIFA Club World Cup, with Group C consisting of Bayern Munich, Benfica, Boca Juniors, and Auckland City, and Group D including Chelsea, Flamengo, Esperance Sportive de Tunisie, and a replacement (possibly Los Angeles FC or Club America based on Article 1). The question specifies that the round of 16 matchup is between the runner-up of Group D and the winner of Group C. Without the actual group stage results, I must make an educated guess based on team strengths. Bayern Munich is a strong candidate for the winner of Group C, as they are the defending champions and one of the top teams. Chelsea, in Group D, could plausibly be the runner-up, given their history, but this is uncertain. The replacement team adds further uncertainty, as it could affect group dynamics."

- **Analysis:** The model selects the more prominent team (Bayern Munich) from Group C rather than correctly identifying which team would actually win the group. It uses team prominence ("defending champions," "top teams") as a proxy for group stage results, ignoring that the question requires specific match outcomes that aren't in the articles.

**4. Incorrect Pattern Matching and Inference**   The model infers answers from similar but not directly relevant information, applying patterns from one context to another inappropriately.

**Example 1:** "What will be the dream main event fight pairing at the UFC card held at the White House to celebrate America's 250th birthday?"

- **Correct answer:** Aspinall vs. Jones

- **Model answer:** Jon Jones vs. Ciryl Gane (probability: 0.1)

- **Model's reasoning:** "The retrieved articles do not directly reference the specific UFC event at the White House for America's 250th birthday, as they focus on unrelated events like UFC 314, UFC 285, and UFC 300. However, Dana White's discussions in Articles 2 and 5 highlight high-profile fights, particularly Jon Jones vs. Ciryl Gane, which is described as a major, sellout event. Given the lack of direct information, I am inferring that this could be a candidate for the "dream main event" based on White's emphasis on big fights, but this is highly speculative due to the time gap and absence of relevant details for the 2026 event. Thus, my best guess is Jon Jones vs. Ciryl Gane."

- **Analysis:** The model incorrectly assumes that a high-profile fight mentioned in articles about other events (UFC 285, UFC 300) will be the main event for the White House card. It applies a pattern ("high-profile fights are main events") without evidence that this specific fight applies to this specific event. The low confidence (0.1) acknowledges speculation but doesn't prevent the incorrect inference.

