# OpenReview forum: "Scaling Open-Ended Reasoning to Predict the Future"
_ICLR.cc/2026/Conference — Submitted to ICLR 2026_

### Official Review · Reviewer_NH88 · 2025-10-29

**Soundness:** 2
**Presentation:** 2
**Contribution:** 2
**Rating:** 2
**Confidence:** 3

**Summary:**

This paper proposes a training framework for open-ended future event forecasting using LLMs. However, despite this ambition, the degree to which the system advances open-ended reasoning for future prediction is not fully demonstrated by the current experiments.

**Strengths:**

- The motivation is clear and timely, targeting the critical but underexplored problem of evaluating models’ ability to forecast real future events.
- The paper contributes a valuable news-based dataset, enabling scalable research on forecasting with LLMs.
- The authors explicitly mitigate data leakage by using offline news, detecting answer leakage in prompts, and rewriting or filtering problematic questions.

**Weaknesses:**

- The SFT + RL training configuration is presented as essential. Still, the experiments do not isolate their respective contributions to reasoning improvements.
- While the task is framed as “open-ended,” most answers are short named entities, and evaluation relies on semantic matching rather than deeper causal understanding.
- Long-term predictive claims depend only on proxy consistency metrics, lacking results on resolved events that truly challenge foresight over extended horizons.

**Questions:**

1. Can the authors provide more fine-grained evidence of reasoning improvement, such as causal inference quality, evidence attribution, or handling of low-signal future scenarios, rather than aggregate accuracy and calibration alone?

2. What distinct roles do SFT and RL play in improving forecasting behavior? Ablations or qualitative analyses would help clarify whether models genuinely anticipate future outcomes rather than react to available information.

3. How do the authors justify the “open-ended” nature of the task, given that answers are primarily short entity names? Could evaluation incorporate more structurally complex future events?

---

> ### Author Response · Authors · 2025-11-20
> **We show improvements from both SFT, and then RL, across all standard LLM forecasting metrics**
>
> Thank you for your comments.
>
> > [W1, Q2] The SFT + RL training configuration is presented as essential. Still, the experiments do not isolate their respective contributions to reasoning improvements. … What distinct roles do SFT and RL play in improving forecasting behaviour?
> >
>
> Figure 7 already shows the change in performance from only doing SFT (grey dots, labeled with a “-sft” suffix), doing SFT+RL (purple dots, labeled with OpenForecaster). We see improvements from both stages, SFT, and RL, with the RL improvements being more in absolute magnitude than SFT, for both brier score and accuracy. We have clarified this in `L453.`
>
> > [W2, Q3] While the task is framed as “open-ended,” most answers are short named entities, and evaluation relies on semantic matching rather than deeper causal understanding. … Could evaluation incorporate more structurally complex future events?
> >
>
> As discussed in `L139-150`, our task is open-ended in the sense that the forecaster does not have to output a structured answer (e.g. number), or select among a few pre-defined choices (e.g. yes/no, or MCQ). Of course, the task could be “more open-ended” in the sense of underspecified questions like “How would US economic policy change in 2026?” but it is unclear how to grade responses to such high levels of open-endedness (thats a good direction for future work). We believe our work takes a valuable step towards making the forecasting task more open-ended, to the extent that the current frontier of automated grading tools allow (see the work on “Answer Matching” by Chandak et al. 2025 cited in the paper).
>
> Do you have a concrete evaluation/benchmark in mind when they say “deeper causal understanding” and “more structurally complex”? We are happy to add those results.
>
> > [W3] Long-term predictive claims depend only on proxy consistency metrics, lacking results on resolved events that truly challenge foresight over extended horizons.
> >
>
> It is not possible to evaluate long-term predictions until they resolve later in the future. The current best methodology we are aware of is the arbitrage metrics proposed in ICLR 2025 Oral paper “Consistency Checks for Language Model Forecasters” by Paleka et al., which we use for our current results, showing large improvements across metrics (see Appendix B.2 for a detailed breakdown). If you have any alternate concrete evaluation/benchmark in mind, we are happy to test it. Note that “long-term predictive” was not a core claim of the paper, as we are well aware this is hard to test.
>
> > [Q1] Can the authors provide more fine-grained evidence of reasoning improvement, such as causal inference quality, evidence attribution, or handling of low-signal future scenarios, rather than aggregate accuracy and calibration alone?
> >
> We are not sure what you mean by “fine-grained evidence”, “causal inference quality”, “evidence attribution”, and “low-signal future scenarios”. If you have any specific evaluations or benchmarks in mind, please do let us know, as we would be happy to try them! Overall, we use standard evaluation metrics in LLM forecasting: accuracy, calibration, consistency for long-term predictions, and show improvements across all of them.

---

> > ### Comment · Reviewer_NH88 · 2025-11-21
> >
> > Thank you for your clarifications on weaknesses 1, 2, and 3. They addressed my concerns to some extent, and I will consider raising the score.
> >
> > Regarding "deeper causal understanding" and "more structurally complex?" in Q1, I would like to restate the core intention more precisely.
> >
> > For open-ended forecasting questions, the model uses underlying reasoning traces or thinking processes (involving causal relationships between retrieved information and the predicted outcome) to generate its prediction. Figure 1 mentions using Grok-3-Mini reasoning traces for SFT, but I was unable to find examples of these traces. Thus, beyond accuracy and the Brier score, I wonder if there is any qualitative evidence showing how the model’s reasoning improves at each training stage. For example, does SFT enhance the use of retrieved evidence? Could the authors provide several illustrative examples?

---

> > > ### Author Response · Authors · 2025-11-27
> > > **Illustrative examples in new Appendix F**
> > >
> > > Thank you for your clarifications!
> > >
> > > We have now provided 3 examples of how the model reasoning evolves during training. We provide the model's own summarized reasoning (after the <think> </think> tags), as the full reasoning trace was too long and confusing to track across checkpoints.
> > >
> > > We see clear signs of the model reasoning about the retrieved information, including the article dates and resolution date. It also reasons about what information it has, and what information is missing, and starts hedging its guesses more as training goes on, realizing its uncertainty. Anecdotally, its reasoning becomes more clearer as training goes on. Later in training, we sometimes even see it explicitly reasoning about the brier score (since its included in the prompt it will be used), and realize it should report its confidence "honestly".
> > >
> > > We hope this clarifies your concern, and thanks for this suggestion, as it improves our paper!

---

> > > > ### Comment · Reviewer_NH88 · 2025-11-27
> > > >
> > > > I appreciate that the authors have included qualitative evidence in the revised version. Event prediction, to some extent, is basically a complex reasoning process, even when the outputs are quantitative ( such as confidence levels or binary indicators). For open-ended questions, the reasoning quality is particularly important, as they depend on timely data and contextual understanding. Such qualitative analysis is valuable. I will raise my score, and my major concerns have been addressed.

---

### Official Review · Reviewer_17m7 · 2025-10-31

**Soundness:** 3
**Presentation:** 2
**Contribution:** 3
**Rating:** 6
**Confidence:** 3

**Summary:**

This paper targets open-ended, probability-bearing forecasting (free-form short answer + confidence) and trains/evaluates language models for future events under strict anti-leakage constraints. The authors propose an automated pipeline, OpenForesight, which continuously synthesizes forecasting questions with explicit resolution dates and evaluation criteria from time-stamped offline news; a multi-stage filtering process yields a high-quality training set.

**Strengths:**

Practical problem: Open-ended prediction with probability output is of clear real-world value.

Scalable data synthesis: From hundreds of thousands of articles to ~60k high-precision samples via multi-stage filtering (validity → best-candidate selection → leakage cleaning → answer-type control), maintaining quality and breadth.

Training recipe that balances goals: SFT warm-start expands solution diversity/ceiling; the joint reward avoids the “accuracy-only hurts calibration / calibration-only suppresses exploration” trade-off.

Time-aware retrieval design (only pre-resolution evidence).

Comprehensive evaluation coverage.

Clear figures and pipeline presentation.

**Weaknesses:**

Compute/cost transparency is limited: missing token counts, steps, GPU-hours for SFT/RL, and end-to-end latency (including retrieval).

No human/market baselines: comparisons to aggregated human forecasters or prediction-market probabilities are absent.

Source/language bias risk: test set from five English outlets—please assess topic/region balance and consider multilingual evaluation.

Long-horizon behavior unclear: what happens for resolutions ≥6/12 months?

Safety: unclear whether sensitive domains (policy/health) receive safety review or filtering.

Open-sourcing: currently a promise without links to released artifacts.

Error analysis: add failure-mode attribution and difficulty stratification of the dataset.

Live evaluation: while I appreciate the anti-leakage design, live evaluation [1] would better reflect deployment and compare with stronger model/agent baselines with no-leak settings.

[1] FutureX: An Advanced Live Benchmark for LLM Agents in Future Prediction

**Questions:**

Please check weakness

---

> ### Author Response · Authors · 2025-11-25
> **We have incorporated your suggestions, thanks!**
>
> Thank you for recognizing our work as being comprehensive, scalable, and having real-world value. We now incorporate your suggestions to further improve our work.
>
> > Live evaluation: while I appreciate the anti-leakage design, live evaluation [1] would better reflect deployment and compare with stronger model/agent baselines with no-leak settings.
> >
>
> We tested our models on questions from FutureX [1], you can see the results in Figure 7b of the updated PDF. OpenForecaster 8B surpassed all other open-weights models we evaluated, including GPT-OSS-120B and DeepSeek-R1. Thanks for the suggestion!
>
> > No human/market baselines: comparisons to aggregated human forecasters or prediction-market probabilities are absent.
> >
>
> Since we finished model development in September, we performed a simulated live test over the last 2 months, with quality filtering on questions similar to Halawi et al (2024). Note that on metaculus, our retrieval system (offline news from 1 month before) leads to an unfair comparisons with humans, as the questions benefit a lot from short-horizon information right before the event resolves. As a result, all models we tested (including OSS 120B and DeepSeek R1) have equal or worse brier than a random predictor (0.25), while humans (who can update their predictions right up till resolution) have an aggregate brier of 0.13. In the new Figure 8 in our paper, we show our training improves an 8B model to achieve better brier score than Qwen3-235-A22B and GPT-OSS-20B.
>
> Please refer to Section 6 of our updated paper for details.
>
> > Compute/cost transparency is limited: missing token counts, steps, GPU-hours for SFT/RL, and end-to-end latency (including retrieval).
> >
>
> We have now added full details about token counts, cost etc. in Appendix F. We used 1 node of 8 H100 GPUs for training. For SFT, fine-tuning `Qwen3-8B` for 3 epochs took 5 hours, corresponding to roughly 40 H100 GPU-hours. Our final RL training run used a single node of 8 H100 GPUs, training on 64,284 samples (roughly 26M input prompt tokens), for 5 epochs, 1300 optimization steps, for a total of around 1,000 H100 GPU hours. Across the full development cycle, including ablations, we estimate using 20,000 H100 GPU hours for this work.
>
> Our curated **OpenForesight** training dataset contains 64,284 samples. The average sample corresponds to roughly 400 tokens under the Qwen3 tokenizer for the question text, yielding approximately **26M** (million) prompt tokens in total. Since our retrieval was offline, the latency is only governed by model output tokens. The model outputs on average 600 tokens, though this varies based on the difficulty of the sample.
>
> > Source/language bias risk: test set from five English outlets—please assess topic/region balance and consider multilingual evaluation.
> >
>
> Indeed, we currently work on english-only data as the goal of this paper was to demonstrate scalable training for forecasting is possible! We look forward to extending training to more languages in the future.
>
> > Long-horizon behavior unclear: what happens for resolutions ≥6/12 months?
> >
>
> The ICLR 2025 Oral paper “Consistency Checks for Language Model Forecasters” (https://openreview.net/forum?id=r5IXBlTCGc) proposes arbitrage metrics to evaluate long-term forecasts up to 2028. On their data and metrics, we report an aggregate of 19% improvement in `L465` with more details in Appendix B.2. This indicates that our training makes models long-term forecasts more consistent.
>
> Note that it is not possible to perform evaluations of ground-truth outcomes of that long a horizon. The training dataset we constructed has questions until March 2025 and even the Qwen3 models we built on were released on April 2025. Questions with more than 6 months horizon have thus not yet resolved to be used for evaluation. We provide monthly accuracy and brier score in Figure 13,14 respectively.
>
> > Open-sourcing: currently a promise without links to released artifacts.
> >
>
> The dataset and model weights exceed the file upload size allowed on OpenReview. We cannot share these artefacts here yet to maintain anonymity as per ICLR guidelines. We will release all artefacts upon publication.
>
> > Error analysis: add failure-mode attribution and difficulty stratification of the dataset.
> >
>
> We provided this in Appendix D, where we find that while our training leads to large improvements (11.6% accuracy) on world-events, it has no significant effect on sports forecasting. We have now also added Appendix G with some qualitative analysis of failures of OpenForecaster 8B.
>
> Your suggestions have greatly helped us improve our paper. We hope this increases your support for our work, and are happy to discuss any further questions.

---

> > ### Author Response · Authors · 2025-11-28
> >
> > Thanks again for your feedback. We made a strong effort to address all your points, including substantial edits to our draft, and we would greatly appreciate it if you would consider increasing your score accordingly. Do you have any other questions?

---

### Official Review · Reviewer_c7uV · 2025-11-01

**Soundness:** 2
**Presentation:** 2
**Contribution:** 2
**Rating:** 4
**Confidence:** 4

**Summary:**

This paper presents a system for training language models to be better at open-ended forecasting. The authors first create OpenForesight, a large-scale training dataset of over 60,000 open-ended questions. This dataset is generated automatically from a static, offline news corpus to prevent data leakage, and then put through a careful, multi-stage curation pipeline to ensure data quality and remove noisy signals. The authors then train Qwen3 models (4B and 8B) using a combination of retrieval, a supervised fine-tuning (SFT) warm-up, and reinforcement learning (RL). A key part of their method is a novel RL reward function that combines both accuracy and the Brier score. The paper demonstrates through ablations that both their data curation recipe and their specific reward function are essential for success. Their final 8B model, OpenForecaster, is shown to match or exceed the performance (specifically in Brier score) of much larger proprietary models on a held-out test set of future events.

**Strengths:**

The paper tackles a valuable and difficult problem: scalable, open-ended forecasting.

It presents a complete, end-to-end system, meticulously validating each component choice, from data curation to the final RL reward.

The data generation and curation pipeline is fully-automated and scalable, and it thoughtfully designed to avoid common pitfalls like data leakage (by using an offline corpus) and self-preference bias (by using different models for generation and filtering).

The ablation studies are a significant strength. The paper empirically proves that its specific data curation recipe is necessary and that its novel 'Accuracy + Brier' reward function is superior to simpler, more common rewards .

The evaluation is rigorous. It uses a held-out test set derived from different news sources than the training set, testing for generalization.

The final result is interesting a specialized, small, open-weight model is shown to be competitive with much larger, general-purpose proprietary models on this specific task.

The authors also demonstrate that the calibration improvements from their training generalize to other standard downstream benchmarks, which strengthens their claims.

Overall I like this paper's focus on forecasting task as I am interested in this shift from fix-known knowledge evaluation to forecasting. I think this paper is a timely paper.

**Weaknesses:**

The paper's novelty is primarily in the systematic combination of existing techniques which makes it hard to pinpoint the exact contribution

The experiments on specialized models are confined to a single model family, Qwen3. This limited scope makes it unclear how dependent the results are on the base Qwen3 model architecture and whether the pipeline and 'Accuracy + Brier' reward would be equally effective if applied to other popular open-weight models. How dependent are the results on the base Qwen3 model architecture? Would the pipeline and 'Accuracy + Brier' reward be equally effective if applied to a Llama or Mistral-family model?

The paper's methodology makes it difficult to disentangle the performance contributions of its two main training stages. It is unclear how much of the final boost comes from the SFT distillation of Grok-3-mini's reasoning versus the new forecasting skill learned during the RL phase on the OpenForesight dataset. Do you have any experiments/results on this?

The reliance on LLMs for the entire curation pipeline might introduce its own set of systematic, unmeasured biases. It is possible that this automated pipeline is systematically filtering out harder, more ambiguous, or more complex forecasting questions, thereby creating a "clean" but simplified version of the real-world task. Basically, the data curation recipe is shown to be effective, but is it possible that this automated pipeline is systematically filtering out harder, more ambiguous, or more complex forecasting questions, thereby creating a "clean" but simplified version of the real-world task?

The training and evaluation setup may obscure the true generalizability of the model. The training set is heavily dominated by two news sources, while the test set uses five completely different ones. This domain mismatch isn't analyzed, making it hard to understand what generalizable skills were learned. This is compounded by the dataset's inherent biases, such as being sourced only from news and the potential for "late reporting" bias.

**Questions:**

See weaknesses.

---

> ### Author Response · Authors · 2025-11-25
> **Improvements Across Model Families, and External Benchmarks**
>
> We are glad you recognize the timeliness of studying the difficult, but valuable problem of forecasting, and the strength of our experiments. We are happy to address your questions below.
>
> > The experiments on specialized models are confined to a single model family, Qwen3.
> >
>
> We repeat the experiments with Llama 3.1 8b, Llama 3.2 3b and Gemma 3 4b (instruct version of each model) training purely with RL (no SFT) and find consistent results: RL training using our dataset, with the accuracy + brier reward leads to large improvements in both calibration and accuracy. The improvements are huge for Llama models with Llama-3.1-8B even surpassing few larger models like GPT-OSS-20B and Qwen3-235B-A22B just with RL training and no SFT. In accuracy, it comes close to our final model based on the Qwen3 backbone but falls a bit short on brier score.
>
> | Model              | Initial Accuracy / Brier | Post-RL Accuracy / Brier |
> |--------------------|--------------------------|---------------------------|
> | llama-3.2-3b       | 4.5 / -0.015             | 41.8 / 0.218              |
> | gemma-3-4b         | 34.1 / -0.173            | 41.1 / 0.157              |
> | llama-3.1-8b       | 22.1 / 0.007             | 47.7 / 0.266              |
> | Qwen3-8B  | 42.2 / 0.214              | 48.8 / 0.33                        |
> | gpt-oss-20b        | 45.9 / 0.233             | –                         |
> | qwen3-235b-a22b    | 45.7 / 0.266             | –                         |
>
>
> Here, for both accuracy and brier, higher is better. Thus, our data and training methodology works well across diverse model families.
>
> > It is unclear how much of the final boost comes from the SFT distillation of Grok-3-mini's reasoning versus the new forecasting skill learned during the RL phase on the OpenForesight dataset.
> >
>
> Note that the questions for SFT are also sourced from our OpenForesight dataset. Figure 7a shows the change in performance from only doing SFT (grey dots, labeled with a “-sft” suffix), doing SFT+RL (purple dots, labeled with OpenForecaster). We see improvements from both stages, SFT, and RL, with the RL improvements being more in absolute magnitude than SFT, for both brier score and accuracy. We have clarified this in `L462.` We also hope our above results on Llama-3.1-8B convince you of the significant improvement achievable via RL alone.
>
> > is it possible that this automated pipeline is systematically filtering out harder, more ambiguous, or more complex forecasting questions, thereby creating a "clean" but simplified version of the real-world task?
> >
>
> Yes, we do agree that LLM based generation and filtering could lead to training data biases. At present, we believe the results showing improvements across benchmarks (see Figure 7b for FutureX, an external benchmark) demonstrate that such an automated pipeline is still valuable. We hope to investigate and fix potential data biases more in the future.
>
> > The training set is heavily dominated by two news sources, while the test set uses five completely different ones. This domain mismatch isn't analyzed, making it hard to understand what generalizable skills were learned.
> >
>
> We think the “domain mismatch” is a feature, not a bug, and we have added the rationale behind it in `L426-427`. Specifically, we tested on different domains precisely to ensure that the learnt forecasting skills are generalizable, and not news source specific distributional biases. This was previously also mentioned in the Introduction (`L80`).
>
> We would also like to inform you that we have run our models even on external benchmarks like FutureX and Metaculus (tested over questions from last 2 months) and we see improvements there as well. Please see Figures 7b and 8 in Section 6 of our updated paper.
>
> > This is compounded by the dataset's inherent biases, such as being sourced only from news and the potential for "late reporting" bias.
> >
>
> We agree that news as a whole has some distributional biases, like not reporting scientific breakthroughs in time etc. In principle, our methodology could be extended to other source documents, like scientific papers. Currently, this is outside the scope of our paper, but we look forward to investigating this in the future. Note that all claims we make in the paper are about *relative* model performance, so these data distribution biases apply across models and should not affect the validity of our claims.
>
> > The paper's novelty is primarily in the systematic combination of existing techniques which makes it hard to pinpoint the exact contribution
> >
>
> Our precise novel contribution is showing how to generate open-ended forecasting training data at scale, which we will release upon publication. This has been a critical bottleneck in the area, leading to existing papers only being focused on evaluating out-of-the-box LLMs.
>
> We hope our clarifications increase your support for our work, and are happy to discuss further.

---

> > ### Author Response · Authors · 2025-11-28
> >
> > Thanks again for your feedback. We made a strong effort to address all your points, including substantial edits to our draft, and we would greatly appreciate it if you would consider increasing your score accordingly. Do you have any other questions?

---

### Official Review · Reviewer_D68d · 2025-11-02

**Soundness:** 3
**Presentation:** 3
**Contribution:** 3
**Rating:** 4
**Confidence:** 4

**Summary:**

This work introduces OpenForesight, a dataset of 60,000 forecasting questions automatically generated from global news articles using the CommonCrawl News corpus and OpenForecaster models, which are Qwen3-4B and 8B trained using SFT on Grok-3-mini reasoning traces, followed by GRPO using accuracy + Brier score as reward. The OpenForecaster 8B model significantly outperforms its base model and matches the Brier score of much larger models, such as gpt-oss-120B, on a held-out test set. The paper also demonstrates that the calibration improvements generalize to standard benchmarks, such as MMLU-Pro and GPQA.

**Strengths:**

1. OpenForesight generation is a scalable, automated pipeline for generating open-ended forecasting questions.
2. The ablation studies provide a thorough validation of design choices like filtering, reward, and retrieval.

**Weaknesses:**

1. The data and models can be heavily skewed towards topics news media covers. Might perform better on domains like politics, etc, but not on long-term cultural shifts, scientific breakthroughs, etc.
2. The filtering pipeline removes ~90% of generated questions. This makes the generation process very expensive and requires further analysis on why the generation model fails so frequently. Will in-context learning help?
3. The paper doesn't justify why numeric answers are filtered out beyond avoiding "vague" responses.
4. The paper shows that training improves forecasting, but provides limited insight into reasoning traces after GRPO and how they hey evolve during training.

**Questions:**

1. Have you considered improving the initial generation prompt or using few-shot examples to increase the acceptance rate of data generation?
2. Do you have any measure of question difficulty?
3. What are the most common failure modes? Are there systematic biases (e.g., toward countries, types of events, recency bias)?
4. Have you tested the system on live forecasting platforms like Metaculus?

---

> ### Author Response · Authors · 2025-11-25
> **Author Response 1/2**
>
> We are glad you liked our core contribution—scalable data generation—and found the ablations thorough. Below we address your questions and concerns.
>
> ## Forecasting model performance
>
> > [Q4] Have you tested the system on live forecasting platforms like Metaculus?
> >
>
> Yes, we tested the model checkpoint (which finished training in September, right after ICLR deadline) over the last 2 months on Metaculus. After applying filtering methods for question quality like Halawi et al. (2024), we found that our training improves forecasts quite a lot. We provide details in Section 6 (Figure 8). Our 8B model surpasses larger models like Qwen3-235-A22B and GPT-OSS-20B. However, all models are worse than the baseline brier score of 0.25, possibly due to lack of the right retrieved documents for answering Metaculus questions.
>
> So we also tested the model on new resolved questions from FutureX benchmark (https://huggingface.co/datasets/futurex-ai/Futurex-Past) and found our 8B model surpasses even the strongest open-weights models like GPT-OSS-120B (Fig. 7). We hope this additional evaluation shows the strength of our dataset and training method, in improving forecasting on both open-ended questions as in our test set, and binary/MCQ style questions in external benchmarks.
>
> > [W4] The paper shows that training improves forecasting, but provides limited insight into reasoning traces after GRPO and how they hey evolve during training.
> >
>
> The most notable change we see is that the model more carefully reasons about uncertainty, and on questions where it is wrong, hedges its guess more over the course of training. We see that the model would earlier instantly jump to an answer (based on its own knowledge) but over the course of training, it learns to better use the articles provided in prompt. Further, it also seems to use more relevant world knowledge from its existing parametric knowledge as well, but this would require more investigation.
>
> We are not sure how to illustrate this, as the reasoning traces are quite long (2 pages on average). Would it be okay for you if we show this on our new Llama 3.1 8B training run, in response to reviewer C7uv? For this model, we see particularly large improvements from training, due to its poor initial performance. Since the reasoning (chain of thought) traces are shorter here, and improvements larger, the evolution during training may be more visible. Please let us know if this would work, and then we can add these to the PDF as well.
>
> > [Q3] What are the most common failure modes? Are there systematic biases (e.g., toward countries, types of events, recency bias)?
> >
>
> We provided this analysis in Appendix D. We find that our model does not improve much on sports related events, but improves significantly on world events. It regresses a bit on location related questions (starting with “where”), but improves on questions starting with “what”). While we expect there to be some country-level bias purely from the fact that we trained on English-only articles, we do not think this is a fundamental property of our methodology. In the future, training on questions sourced from a larger, more diverse set of documents would in principle mitigate this.
>
> We have now also added Appendix G, with some more qualitative analysis on failure modes. Thanks for the suggestion.
>
> Continued below...

---

> ### Author Response · Authors · 2025-11-25
> **Author Response 2/2**
>
> ## Data generation
>
> > [W2/Q1] The filtering pipeline removes ~90% of generated questions. This makes the generation process very expensive … Have you considered improving the initial generation prompt or using few-shot examples to increase the acceptance rate of data generation?
> >
>
> Great question! Our initial question generation prompt is provided on Page 22-27. You will notice its quite long. It includes detailed guidelines, with examples to illustrate the guidelines, and also a full concrete example! We iterated on this prompt over months after many rounds of prompt updates and manual inspection. We agree that ideally needing less filtering would make the training data generation process even more scalable. While a 90% rejection rate in questions can seem high, it arises from the following factors:
>
> (1) We wanted to maintain a high bar for quality.
>
> (2) Unambiguous forecasting questions are deceptively challenging to create, for example, see this [list of common issues](https://rethinkpriorities.org/research-area/types-of-specification-problems-in-forecasting/). There are many subtle constraints that must be followed which we listed in our initial prompt. Following all these constraints is a complex constraint satisfaction problem, and the validation steps help us filter out initial generations that do not follow important constraints.
>
> (3) Even in prior work like Halawi et al. (2024) which sourced human created questions from platforms like Polymarket and Metaculus, they had to apply extensive filtering to remove low quality questions. For eg in Halawi et al they started with 33,664 binary questions which they filtered down to 5,516 (roughly 6x reduction in size). Considering that our questions are open-ended, which requires more care to ensure there’s no ambiguity in the resolution criteria, and extracted by an LLM from the source documents, we think a 10x reduction in size from the initial generation is quite decent.
>
> (4) We used a non-reasoning LLM (DeepSeek v3) to generate the initial train set questions, precisely to mitigate costs. The test set questions were actually created with o4-mini (high) and there our filtering rate was much lower (~50% filtered instead of 90%). We could have used o4-mini-high to increase the retaining rate of our pipeline but the cost of o4-mini-high is 10x that of DeepSeek v3. We expect that both the rejection rate and cost per generation would go down in the future as capability per unit cost ratios improve.
>
> (5) The question rejection rate is also quite dependent on the source articles, as we found some sources systematically worse for creating questions.
>
> Note that the overall dataset creation process costed us (approx.) 3000USD with training set costing 2200 (using DeepSeek-v3) while creating the test set costing 750 (using o4-mini-high). While this amount is significant (which also highlights the importance of us eventually open-sourcing this training data), note that if we hired humans, the estimated cost would be much higher. For example, [Mercor is currently hiring forecasters at $105-245 per hour](https://work.mercor.com/explore?listingId=list_AAABmkAAOfV2o3kz69hKOLIU). Even if we ambitiously estimated they could make 20 forecasting questions per hour, it would cost upwards of 300,000$ (>100x more) to create the 60,000 questions in our training data.
>
> > [W1] The data and models can be heavily skewed towards topics news media covers. Might perform better on domains like politics, etc, but not on long-term cultural shifts, scientific breakthroughs, etc.
> >
>
> This is true for our current dataset and models. In principle, one could extend our methodology to other source documents which capture long-term cultural shifts and scientific breakthroughs, and we look forward to investigating this in future work.
>
> > [W3] The paper doesn't justify why numeric answers are filtered out beyond avoiding "vague" responses.
> >
>
> We filter numeric questions to manage the scope of this study. It is unclear what is the best way to aggregate numeric prediction performance with the more discrete predictions (accuracy / brier score). One would also need a different metric like relative error or mean squared error to score numeric predictions, with interval confidence. We hope to incorporate numeric questions and predictions in the future.
>
> > [Q2] Do you have a measure for question difficulty
> >
>
> In principle, the aggregate absolute accuracy across models could be used as a measure of question difficulty. However, we don’t go beyond this for now as we are interested in *relative* comparisons between models. Note that the difficulty of questions for a model would also vary based on the external information retrieved. On our training set, without any retrieval, Qwen3-8B has ~18% accuracy. Post retrieval, it has around ~30% accuracy.
>
> Thanks for these questions and suggestions. They have helped improve our paper, and we hope this increases your support for our work.

---

> ### Comment · Reviewer_D68d · 2025-11-27
> **Response to Authors Rebuttal**
>
> I thank the authors for their response. I still feel the scope of this work is very limited. I'm keeping my score the same.

---

### Author Response · Authors · 2025-12-03
**Final Author Remarks**

We thank the reviewers for their constructive feedback which helped further improve our paper during the rebuttal phase.

We appreciate that the reviewers recognize our work on scalable training for open-ended LLM forecasting is *“clear and timely, targeting a critical but underexplored problem”* (`Reviewer NH88`), with `Reviewers c7uV, 17m7` also highlighting that the problem we study is *“challenging”* and has *“practical, … real-world value”.* We are glad they found our experiments *“thorough”* and ablations of design choices *“meticulous”* (`Reviewers D68d, C7uv`). `Reviewers c7uV, 17m7, NH88` appreciated the *“rigour”* and *“comprehensive coverage”* of our evaluations, highlighting how we *“explicitly mitigate data leakage by using offline news, detecting answer leakage in prompts, and rewriting or filtering problematic questions.”* As noted by `Reviewers D68d, c7uV`, this results in *“a specialized, small, open-weight model competitive with much larger, general-purpose proprietary models on this specific task”*, and *“calibration improvements generalize to other standard downstream benchmarks”,* demonstrating the success of our proposed methodology for scalable training data generation for open-ended forecasting.

---

We now summarize the improvements we made during the rebuttal phase to address all questions and weaknesses brought up by the reviewers.

| Suggestion | Improvements |
| --- | --- |
| More diverse evaluations (`All Reviewers` ) | We tested our 8B model that finished training in September (on training data until April 2025) on 2 new datasets: `FutureX`, and `resolved questions on Metaculus from the last 2 months`. We see consistent improvements, surpassing larger models like Qwen3-235-A22B and GPT-OSS-20B and becoming competitive with GPT OSS 120B as shown in `Figures 7,8` in the updated PDF. This addresses questions about live-testing and dataset bias, showing our training improves LLM forecasting across datasets and newer events. |
| More model families, SFT vs RL (`Reviewers c7uV, NH88`) | We added Figure 15 showing that RL training (without SFT) on our dataset leads to large improvements for even Llama and Gemma models, in both accuracy and brier score. We clarified in `L462`that for our main model built on top of Qwen3-8B, both phases, SFT on Grok-3-mini reasoning traces and RL, contribute to improvements, with RL leading to more absolute gains. |
| Error and Reasoning Analysis (`Reviewers D68d, 17m7, NH88`) | We expanded our qualitative analysis from the existing Appendix D to include examples of reasoning trace evolution during training (Appendix F), and failure mode analysis (Appendix H). This led to **Reviewer NH88 increasing their score from 2 to 6**, as they were convinced the model better reasons about retrieved evidence, uncertainty, and makes causal inferences. |
| Data generation and cost (`Reviewers D68d, c7uv, 17m7`) | We answered all questions about the data generation process, providing detailed analysis on how it saves significant cost compared to collecting questions from human forecasters, making it more scalable for training. We clarified how testing on events drawn from different news domains than than the ones used in training was a feature of our study to ensure our trained model is not just learning distributional biases, but general forecasting. Finally, we provide a breakdown of compute and data generation costs in Appendix G. Our methodology can be extended to other data formats like numeric, and source documents different from global news in the future.  |

We incorporate all these experiments and clarifications in the updated PDF, with changes colored in $\color{blue}{blue}$. Overall, 3/4 reviewers did not get time to acknowledge our response, so we hope this short summary helps the new AC understand how we addressed all reviewer feedback.

---

### Meta-Review · Area_Chair_dneh · 2026-01-07

**Summary:**

This paper studies open-ended forecasting with LLMs, focusing on predicting future events. The authors propose a pipeline, OpenForesight, which generates open-ended forecasting questions from offline global news articles, applies multi-stage filtering to avoid ambiguity and data leakage, and trains language models using a combination of supervised fine-tuning on reasoning traces and reinforcement learning with an accuracy plus Brier score reward. The resulting model, OpenForecaster-8B, is shown to achieve competitive accuracy, calibration, and consistency compared to much larger proprietary models on held-out future events and external benchmarks. The paper also emphasizes calibration improvements and claims these generalize across standard downstream forecasting benchmarks.

The initial reviewer scores are mixed, with scores of 4, 4, 6, and 2. The authors provided a rebuttal, and during the discussion phase some reviewers explicitly indicated that they would raise their scores. As the Area Chair, I carefully read the paper, all reviewer comments, the authors’ responses, and the ensuing discussion.

**Reviewer Concerns:**

Reviewer D68d raised an important concern regarding dataset and model bias. Specifically, the reviewer pointed out that both the generated data and trained models may be skewed toward topics heavily covered by news media, such as politics or short-term world events, while underrepresenting domains like long-term cultural change or scientific breakthroughs. The authors responded by acknowledging this limitation and framing it largely as future work, also discussing constraints imposed by using offline news corpora and GPT-based generation. While this response is reasonable to an extent, I believe the paper should go further. Given that this work positions itself as introducing a new dataset and research paradigm for forecasting, it is important that the manuscript and supplementary materials include sufficiently rich reasoning traces and qualitative analyses to make the feasibility and limitations of such forecasting explicit. This concern also recurs in later reviews. After discussion, Reviewer D68d maintained the view that the work remains in need of improvement.

Reviewer c7uV focused on the dependence on the Qwen3 model family and on disentangling the effects of the two main training stages, supervised fine-tuning versus reinforcement learning. This reviewer also raised concerns about systematic biases introduced by the fully automated data generation and filtering pipeline. The authors responded with additional experiments on Llama and Gemma models, as well as clarifications on the relative contributions of SFT and RL. From the AC’s perspective, the discussion of inherent dataset bias remains limited. The fact that such biases may affect all models trained on the dataset does not eliminate the need to explicitly analyze and discuss their sources, distributions, and implications. For a paper that proposes a new dataset, these issues should be treated as first-order considerations rather than deferred to future work.

Reviewer 17m7 raised concerns about transparency in computational cost, lack of human or market baselines, safety considerations, and error analysis. The authors responded by adding simulated live testing, reporting compute usage, and providing additional analyses.

Reviewer ZH88 emphasized the need for stronger evidence of reasoning improvement, particularly through detailed qualitative examples, and raised concerns about causal inference quality, attribution of evidence, and handling of first-order signals about future events. The authors responded by adding several illustrative examples of reasoning evolution during training. While these examples are a step in the right direction, I do not find them sufficient. For a paper that claims advances in open-ended reasoning for future prediction, much more careful and systematic discussion of causal reasoning quality, evidence attribution, and failure cases is needed. These aspects are central to the contribution and cannot be convincingly established with a small number of examples.

Although some reviewers raised their scores after discussion, I believe the bar for this type of work should remain high. The paper tackles an interesting and important problem and explores it from a creative angle. However, significant concerns remain unresolved: dataset and topic bias are not sufficiently analyzed; the evidence for genuine improvements in causal and open-ended reasoning remains limited; and several key claims rely heavily on aggregate metrics without enough qualitative grounding. In addition, the writing would benefit from clearer emphasis on limitations and more rigorous justification of the proposed paradigm’s scope and implications.

**Reviewer Scores:**

see above

---

### Decision · Program_Chairs · 2026-01-26

Reject